# Credit Assignment Through Broadcasting a Global Error Vector

**David G. Clark, L.F. Abbott, SueYeon Chung**
Center for Theoretical Neuroscience
Columbia University
New York, NY
{david.clark, lfabbott, sueyeon.chung}@columbia.edu

## Abstract

Backpropagation (BP) uses detailed, unit-specific feedback to train deep neural networks (DNNs) with remarkable success. That biological neural circuits appear to perform credit assignment, but cannot implement BP, implies the existence of other powerful learning algorithms. Here, we explore the extent to which a globally broadcast learning signal, coupled with local weight updates, enables training of DNNs. We present both a learning rule, called global error-vector broadcasting (GEVB), and a class of DNNs, called vectorized nonnegative networks (VNNs), in which this learning rule operates. VNNs have vector-valued units and nonnegative weights past the first layer. The GEVB learning rule generalizes three-factor Hebbian learning, updating each weight by an amount proportional to the inner product of the presynaptic activation and a globally broadcast error vector when the postsynaptic unit is active. We prove that these weight updates are matched in sign to the gradient, enabling accurate credit assignment. Moreover, at initialization, these updates are exactly proportional to the gradient in the limit of infinite network width. GEVB matches the performance of BP in VNNs, and in some cases outperforms direct feedback alignment (DFA) applied in conventional networks. Unlike DFA, GEVB successfully trains convolutional layers. Altogether, our theoretical and empirical results point to a surprisingly powerful role for a global learning signal in training DNNs.

## 1 Introduction

Deep neural networks (DNNs) trained using backpropagation (BP) have achieved breakthroughs on a myriad of tasks [1, 2]. The power of BP lies in its ability to discover intermediate feature representations by following the gradient of a loss function. In a manner analogous to DNN training, synapses in multilayered cortical circuits undergo plasticity that modulates neural activity several synapses downstream to improve performance on behavioral tasks [3, 4]. However, neural circuits cannot implement BP, implying that evolution has found another algorithm, or collection thereof [5]. This observation has motivated biologically plausible alternatives to BP [6, 7].

Credit assignment algorithms based on broadcasting a global learning signal, such as node perturbation [8], are attractive due to their biological plausibility, as the global signal could correspond to a neuromodulator that influences local synaptic plasticity [9, 10]. However, perturbation methods are of little practical relevance as the variance in their gradient estimates is prohibitively large [7, 11]. Instead, recent research has focused on methods that compute or estimate the gradient by transmitting detailed, unit-specific information to hidden layers through top-down feedback. The foremost example of such a method is BP. BP's use of the feedforward weights in the feedback

---

Code accompanying our paper is available at https://github.com/davidclark1/VectorizedNets.

35th Conference on Neural Information Processing Systems (NeurIPS 2021).

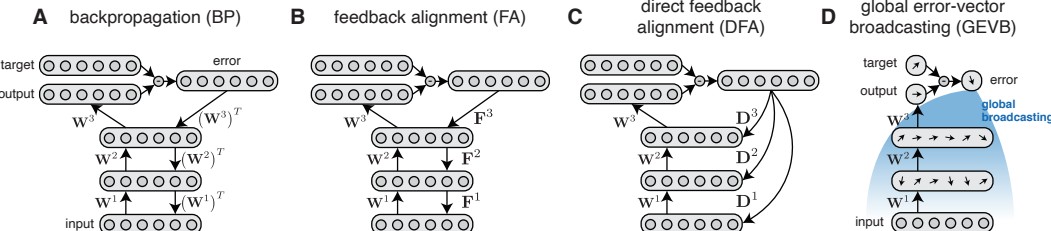

Figure 1: Credit assignment methods. **(A)** Backpropagation (BP) transmits the error vector backwards layer-by-layer using the transposes of the feedforward weights. **(B)** Feedback alignemnt (FA) transmits the error vector backwards layer-by-layer using fixed, random feedback matrices. **(C)** Direct feedback alignment (DFA) delivers the error vector directly to each hidden layer using fixed, random feedback matrices. **(D)** Global error-vector broadcasting (GEVB) conveys the full error vector to all hidden units without unit-specific feedback. Thus, there are no feedback parameters. GEVB operates in networks in which each hidden unit is vector-valued. Each arrow represents a single vector unit.

pathway, a property called weight symmetry, is not biologically plausible – this is the weight transport problem [12]. Another method of this class, feedback alignment (FA), operates identically to BP but uses fixed, random matrices in the feedback pathway, thereby circumventing the weight transport problem [13]. FA inspired a further method called direct feedback alignment (DFA), which delivers random projections of the output error vector directly to hidden units [14, 15]. DFA approaches the performance of BP in certain modern architectures [16], however its performance still lags behind that of BP in many cases [14, 16–19]. A particularly striking shortcoming of DFA is its inability to train convolutional layers [16, 19, 20]. To be implemented in neural circuits, both FA and DFA require a biologically unrealistic "error network" to compute top-down learning signals, though models based on segregated dendrites aim to lift this requirement [21–23].

Here, we show that DNNs can be trained to perform on par with BP by broadcasting a single, global learning signal to all hidden units and applying local, Hebbian-like updates to the weights. Our proposed learning rule, called global error-vector broadcasting (GEVB), is not perturbation-based, but instead distributes information about the output error throughout the network. Unlike DFA, which delivers a unique random projection of the output error vector to each hidden unit, GEVB broadcasts the same, unprojected error vector to all hidden units. Thus, GEVB involves *no unit-specific feedback*. This learning rule operates in a new class of DNNs called vectorized nonnegative networks (VNNs), which have vector-valued units and nonnegative weights past the first layer. While vector-valued units are neuroscientifically speculative (though see Section 7 for possible neural implementations), nonnegative weights are motivated by the fact that cortical projection neurons are excitatory [4]. The GEVB learning rule updates each weight by an amount proportional to the inner product of the presynaptic activation and the global error vector when the postsynaptic unit is active. Three-factor Hebbian plasticity encompasses synaptic update rules that depend on pre- and postsynaptic activity and a global neuromodulatory signal (the third factor) [24, 25]. The GEVB learning rule is therefore a form of three-factor Hebbian plasticity with a vector-valued global third factor [26]. Our experimental results show that this form of global-error learning is surprisingly powerful, performing on par with BP in VNNs and overcoming DFA's inability to train convolutional layers.

## 2 Credit assignment in conventional networks

Consider a fully connected network of $L$ layers with pre-activations $h_i^\ell$ and activations $a_i^\ell$, where $i = 1, \ldots, n_\ell$ and $\ell = 0, \ldots, L$. Let $w_{ij}^\ell$ denote the weights, $b_i^\ell$ the biases, and $\phi$ the pointwise nonlinearity. The forward dynamics are

$$h_i^\ell = \sum_{j=1}^{n_{\ell-1}} w_{ij}^\ell a_j^{\ell-1} + b_i^\ell, \quad a_i^\ell = \phi\left(h_i^\ell\right), \tag{1}$$

with the understanding that $\phi$ is the identity function at the output layer $L$. The network inputs and outputs are $a_i^0$ and $a_i^L$, respectively. We place a loss $\mathcal{L}$ on the output, and define $\partial \mathcal{L}/\partial a_i^L = e_i$, where $e_i$ is the output error vector. On a single training example, a negative-gradient weight update can be

written

$$\Delta w_{ij}^\ell \propto -\delta_i^\ell \phi'\left(h_i^\ell\right) a_j^{\ell-1} \quad \text{where} \quad \delta_i^\ell = \frac{\partial \mathcal{L}}{\partial a_i^\ell} = \sum_{k=1}^{n_L} \frac{\partial a_k^L}{\partial a_i^\ell} e_k. \tag{2}$$

BP, FA, and DFA are different methods of computing or approximating $\delta_i^\ell$. BP computes $\delta_i^\ell$ exactly using the recurrence relation

$$\delta_i^\ell = \sum_{k=1}^{n_{\ell+1}} w_{ki}^{\ell+1} \phi'(h_k^{\ell+1}) \delta_k^{\ell+1} \tag{3}$$

with the initial condition $\delta_i^L = e_i$ (Fig. 1A). FA approximates $\delta_i^\ell$ by performing a backward pass identical to Eq. 3, but using fixed, random feedback weights $f_{ik}^\ell$ in place of the transposed feedforward weights $w_{ki}^\ell$ (Fig. 1B). This backward pass is given by

$$\delta_i^\ell = \sum_{k=1}^{n_{\ell+1}} f_{ik}^{\ell+1} \phi'(h_k^{\ell+1}) \delta_k^{\ell+1} \tag{4}$$

with the initial condition $\delta_i^L = e_i$. Learning occurs under Eq. 4 due to learning-induced partial alignment of $w_{ki}^\ell$ with $f_{ik}^\ell$ [13]. Finally, DFA approximates $\delta_i^\ell$ by setting the derivative $\partial a_k^L / \partial a_i^\ell$ in the formula for $\delta_i^\ell$ in Eq. 2 to a fixed, random matrix $d_{ik}^\ell$ (Fig. 1C). This yields

$$\delta_i^\ell = \sum_{k=1}^{n_L} d_{ik}^\ell e_k. \tag{5}$$

In a similar manner to FA, learning occurs in DFA due to induced partial alignment of $\partial a_k^L / \partial a_i^\ell$ with $d_{ik}^\ell$ [19, 20]. The learning rule we propose further simplifies Eqns. 3, 4, and 5 by removing computation of the quantity analogous to $\delta_i^\ell$ entirely (Section 3.2).

A fundamental obstacle to accurate credit assignment in the absence of weight symmetry is that the sign of the gradient at each weight depends on detailed information about downstream weights to which the feedback pathway does not have access, even in the presence of partial alignment of feedforward and feedback weights. For example, suppose a DNN confidently predicts class $c'$ instead of the target class $c$. Then, the gradient at $w_{ij}^\ell$ depends on $\delta_i^\ell \approx \partial a_{c'}^L / \partial a_i^\ell - \partial a_c^L / \partial a_i^\ell$, which can be positive or negative depending on a difference in the strengths of complex polysynaptic interactions between unit $i$ in layer $\ell$ and units $c$ and $c'$ in layer $L$.

Intriguingly, this obstacle dissolves in networks with scalar output, positive weights past the first layer, and $\phi' > 0$. In this case, we have $\delta_i^\ell = (\partial a_1^L / \partial a_i^\ell) e_1$ with $\partial a_1^L / \partial a_i^\ell > 0$. Thus, as per Eq. 2, the gradient sign at $w_{ij}^\ell$ is equal to the sign of $\phi'(h_i^\ell) a_j^{\ell-1} e_1$. This expression has the form of a three-factor Hebbian rule. This simplification to credit assignment arising from sign-constrained weights in scalar-output networks was observed by both Balduzzi et al. [27], who proposed the Kickback algorithm, and Lechner [18], who used sign-constrained weights in DFA. However, the restriction to scalar-output (e.g., binary classification) networks is highly limiting. The GEVB learning rule, described in the next section, generalizes this observation to vector-output (e.g., multi-way classification) networks.

## 3 Credit assignment in vectorized nonnegative networks

### 3.1 Vectorized nonnegative networks

Here, we introduce VNNs, the class of DNNs in which our proposed learning rule operates. Given a task with output dimension $K$, such as $K$-way classification, each VNN unit is a vector of dimension $K$. With the exception of the first hidden layer, connections obey "vectorized weight sharing" such that each vector unit computes a weighted linear combination of presynaptic vector units. Our notational convention is to use Latin letters to index different vector units and Greek letters to index the components of a single vector unit. Thus, Greek indices always run over $1, \ldots, K$. Boldface variables represent all $K$ components of a vector unit.

For $\ell = 1 \ldots, L$, let $h_{i\mu}^\ell$ denote the pre-activations and $a_{i\mu}^\ell$ the activations. The inputs, $a_i^0$, are not vectorized and hence lack a $\mu$ subscript. The vector units in the first hidden layer compute a

representation of the input using the weights $w^1_{i\mu j}$. For $\ell > 1$, connections obey vectorized weight sharing. Thus, these weights lack a $\mu$ subscript and are denoted by $w^\ell_{ij}$. Crucially, these $\ell > 1$ weights are nonnegative, consistent with excitatory cortical projection neurons. For $\ell = 1, \ldots, L$, the biases are vector-valued and are denoted by $b^\ell_{i\mu}$. Finally, the vector-to-vector nonlinearity is denoted by $\Phi : \mathbb{R}^K \to \mathbb{R}^K$. This function mixes vector components in general. The forward dynamics are

$$
h^\ell_{i\mu} = \begin{cases} \sum\limits_{j=1}^{n_0} w^1_{i\mu j} a^0_j + b^1_{i\mu} & \ell = 1 \\ \sum\limits_{j=1}^{n_{\ell-1}} w^\ell_{ij} a^{\ell-1}_{j\mu} + b^\ell_{i\mu} & \ell > 1 \end{cases} \qquad a^\ell_{i\mu} = \Phi_\mu\left(h^\ell_i\right) \tag{6}
$$

with the understanding that $\Phi$ is the identity function at the output layer $L$. We assume that there is a single vector output unit $a^L_{1\mu}$, denoted by $a^L_\mu$ for brevity, on which we place a loss $\mathcal{L}$. We define $\partial\mathcal{L}/\partial a^L_\mu = e_\mu$, where $e_\mu$ is the output error vector. We will show that our proposed learning rule matches the sign of the gradient when the nonlinearity has the form

$$
\Phi_\mu(\boldsymbol{h}) = G(\boldsymbol{h})h_\mu, \tag{7}
$$

with $G(\boldsymbol{h}) \geq 0$ and piecewise constant. The function $G$ induces a coupling between different vector components. While there are many options for this function, we choose

$$
G(\boldsymbol{h}) = \Theta\left(\boldsymbol{t} \cdot \boldsymbol{h}\right), \tag{8}
$$

where $\Theta$ is the Heaviside step function and $\boldsymbol{t}$ is a gating vector that differs across units (for brevity, we suppress its $i$ and $\mu$ indices in our notation). We sample the gating vectors uniformly over $\{-1, 1\}^K$ at initialization and hold them constant throughout training and inference. In the $K = 1$ case, this choice of $\Phi$ reduces to a ReLU if $t > 0$ since $\Phi(h) = \Theta(th)h = \Theta(h)h = \text{ReLU}(h)$.

While we described the inputs as $n_0$ scalar units with all-to-all connections to the first hidden layer, the inputs can be described equivalently as $Kn_0$ vector units with vectorized weight-shared connections to the first hidden layer, unifying the $\ell = 1$ and $\ell > 1$ cases of the learning rule (Appendix B). We use this convention henceforth.

### 3.2 Global error-vector broadcasting learning rule

The GEVB learning rule works by globally broadcasting the output error vector $e_\mu$ to all hidden units:

$$
\textbf{GEVB:} \quad \Delta w^\ell_{ij} \propto -G\left(\boldsymbol{h}^\ell_i\right) \sum_\mu a^{\ell-1}_{j\mu} e_\mu. \tag{9}
$$

Thus, presynaptic units that are aligned or anti-aligned with the output error vector have their weight onto the postsynaptic unit decreased or increased, respectively, when the postsynaptic unit is active. Note that this learning rule has *no feedback parameters*. The GEVB learning rule can be interpreted as a three-factor Hebbian rule with a vector-valued global third factor $e_\mu$ [24–26]. Our experimental results in Section 6 show that this extremely simple learning rule is surprisingly powerful.

### 3.3 Global error-vector broadcasting matches the sign of the gradient

We now show that by choosing $\Phi$ as in Eq. 7, with $G(\boldsymbol{h}) \geq 0$ and piecewise constant, GEVB weight updates are matched in sign to the gradient. The gradient on a single training example is

$$
\frac{\partial\mathcal{L}}{\partial w^\ell_{ij}} = G(\boldsymbol{h}^\ell_i) \sum_{\mu,\nu} e_\mu \frac{\partial a^L_\mu}{\partial a^\ell_{i\nu}} a^{\ell-1}_{j\nu}. \tag{10}
$$

The $K$-by-$K$ Jacobian $\partial a^L_\mu/\partial a^\ell_{i\nu}$ in Eq. 10 describes how component $\nu$ of vector unit $i$ in layer $\ell$ polysynaptically influences component $\mu$ of the vector output unit. This Jacobian can be computed iteratively in a BP-like manner,

$$
\frac{\partial a^L_\mu}{\partial a^\ell_{i\nu}} = \sum_{k=1}^{n_{\ell+1}} w^{\ell+1}_{ki} G(\boldsymbol{h}^{\ell+1}_k) \frac{\partial a^L_\mu}{\partial a^{\ell+1}_{k\nu}} \tag{11}
$$

with the initial condition $\partial a_\mu^L/\partial a_{1\nu}^L = I_{\mu\nu}$, where $I_{\mu\nu}$ is the identity matrix. These recursions ensure that $\partial a_\mu^L/\partial a_{i\nu}^\ell$ is proportional to $I_{\mu\nu}$ for all $\ell$. Thus, we write

$$\frac{\partial a_\mu^L}{\partial a_{i\nu}^\ell} = \hat{\delta}_i^\ell I_{\mu\nu} \quad \text{where} \quad \hat{\delta}_i^\ell = \frac{\partial a_\mu^L}{\partial a_{i\mu}^\ell}. \tag{12}$$

Substitution of Eq. 12 into Eq. 11 gives

$$\hat{\delta}_i^\ell = \sum_{k=1}^{n_{\ell+1}} w_{ki}^{\ell+1} G(\boldsymbol{h}_k^{\ell+1})\hat{\delta}_k^{\ell+1} \tag{13}$$

with the initial condition $\hat{\delta}_i^L = 1$. Altogether, substituting Eq. 12 into Eq. 10, the gradient is

$$\frac{\partial \mathcal{L}}{\partial w_{ij}^\ell} = \hat{\delta}_i^\ell G(\boldsymbol{h}_i^\ell) \sum_\mu a_{j\mu}^{\ell-1} e_\mu, \tag{14}$$

where $\hat{\delta}_i^\ell$ is backpropagated according to Eq. 13. The nonnegativity of the $\ell > 1$ weights and of $G(\boldsymbol{h})$ ensure that $\hat{\delta}_i^\ell \geq 0$ for all $\ell$. Under a mild additional assumption (Appendix C), we have strict positivity, $\hat{\delta}_i^\ell > 0$. Thus, setting the $\hat{\delta}_i^\ell$ term in Eq. 14 to a positive constant yields an expression with the same sign as the gradient. This is precisely the GEVB learning rule of Eq. 9.

Starting from a general vectorized network, one can derive weight nonnegativity, the required form of the vector nonlinearity, and the GEVB learning rule itself from the requirement that the gradient sign is computable using only pre- and postsynaptic activity and the output error vector (Appendix D).

Recent theoretical work has shown that gradient sign information is sufficient for attaining or improving stochastic gradient descent convergence rates in non-convex optimization [28]. A caveat is that these rates assume that the gradient sign is computed on mini-batches, whereas GEVB matches the gradient sign on individual examples. In Section 6, we show that training VNNs using batched GEVB updates yields performance on par with BP. Whether non-batched GEVB updates are particularly effective by virtue of matching the gradient sign exactly is a question for future study. Finally, we note that prior studies have demonstrated strong performance of a variant of FA, called sign symmetry, in which the feedback matrices have the same sign as the feedforward matrices [29–31]. By contrast, GEVB has no feedback parameters and provides weight updates with the same sign as the gradient. Sign symmetry has no such theoretical guarantee and thus presumably results in a looser approximation of the gradient.

## 4   Gradient alignment beyond sign agreement

The GEVB learning rule approximates the gradient by setting $\hat{\delta}_i^\ell$ in Eq. 14 to a positive constant for all units $i$ in every layer $\ell$. This approximation is accurate if the empirical distribution of $\hat{\delta}_i^\ell$ in layer $\ell$ is tightly concentrated about a positive value, and poor if this distribution is diffuse. The level of concentration can be quantified as the relative standard deviation $r^\ell = \sigma_{\hat{\delta}^\ell}/\mu_{\hat{\delta}^\ell}$, where $\mu_{\hat{\delta}^\ell}$ and $\sigma_{\hat{\delta}^\ell}^2$ denote the mean and variance of the empirical distribution of $\hat{\delta}_i^\ell$ in layer $\ell$. Since $\hat{\delta}_i^\ell > 0$, we have $r^\ell > 0$. When $r^\ell \ll 1$, this distribution is close to a delta function at a positive value and $\hat{\delta}_i^\ell$ is accurately approximated as a positive constant. The relative standard deviation is closely related to the angle $\theta^\ell$ between GEVB weight updates and the gradient according to $\theta^\ell = \tan^{-1} r^\ell$ (see Appendix E for proof). This alignment-angle metric is commonly used to measure the quality of weight updates for learning [13, 19].

Intriguingly, in randomly initialized VNNs, $r^\ell$ scales inversely with $n_\ell$ in sufficiently early layers so that, in the limit of infinite network width, $r^\ell, \theta^\ell \to 0$. Concretely, if the elements of $\boldsymbol{W}^\ell$ are sampled i.i.d. from a distribution with mean $\mu_{w^\ell} > 0$ and variance $\sigma_{w^\ell}^2$, with $\sigma_{w^\ell}/\mu_{w^\ell}$ smaller than order $\sqrt{n_\ell}$, $r^\ell$ obeys the recurrence relation

$$r^\ell = \sqrt{\frac{2}{n_{\ell+1}}} \frac{\sigma_{w^{\ell+1}}}{\mu_{w^{\ell+1}}} \sqrt{1 + (r^{\ell+1})^2} \tag{15}$$

with the initial condition $r^{L-1} = \sigma_{w^L}/\mu_{w^L}$ (see Appendix F for derivation; layer-$L$ GEVB updates are equal to the gradient by definition). Since $\sigma_{w^\ell}/\mu_{w^\ell}$ is smaller than order $\sqrt{n_\ell}$, $r^\ell$ scales inversely with $n_\ell$ after sufficiently many recursions. Our experiments in VNNs used a non-i.i.d. initialization described in Section 5 whose backward-pass behavior is qualitatively similar to using i.i.d. weights with $\sigma_{w^\ell}/\mu_{w^\ell} \sim 1$. Our measurements of GEVB alignment angles at initialization (Section 6; Fig. 2A) were in agreement with Eq. 15. Specifically, in layer $L-1$, we observed $\theta^{L-1} \sim 45°$, corresponding to $r^{L-1} \sim 1$; in layers $\ell < L-1$, we observed small $\theta^\ell$, corresponding to $r^\ell \sim 1/\sqrt{n_{\ell+1}}$.

The limit of infinite network width has been shown to dramatically simplify the dynamics of gradient descent in DNNs, an idea embodied in the Neural Tangent Kernel (NTK) [32, 33]. Our finding that credit assignment is simplified in the same limit suggests a potentially fruitful connection between the NTK regime and biologically plausible DNN training.

## 5  Initializing nonnegative networks with ON/OFF cells

GEVB can be applied in networks initialized with zero weights, a method sometimes used with FA and DFA. However, BP is incompatible with zero initialization as the gradient vanishes. To enable fair comparisons between GEVB and BP, we used the same nonzero initialization for both training methods. Random nonnegative initializations have the added benefit of placing VNNs in a regime in which GEVB weight updates are highly gradient-aligned (Section 4). Unfortunately, nonnegative i.i.d. initializations are unsuitable for initializing DNNs as they tend to produce exploding forward passes. Specifically, if the weights in layer $\ell$ are sampled i.i.d. from a distribution with mean $\sim 1$ and variance $\sim 1/n_{\ell-1}$, the weight matrix has an outlier eigenvalue of size $\sim \sqrt{n_{\ell-1}}$, and the projection of the pre-activations onto this mode grows across layers. Here, we present a nonnegative initialization for VNNs based on ON/OFF cells that yields well-behaved forward propagation.

First, we group the hidden units into pairs with equal-and-opposite gating vectors $\boldsymbol{t}$, defining the structure of ON and OFF cells. We then initialize the weights such that the units in each pair have equal-and-opposite activations. The subnetwork of ON cells has the same activations as a network of half the size with i.i.d. mixed-sign weights, and thus the ON/OFF initialization exhibits well-behaved forward propagation insofar as the underlying mixed-sign initialization does. To initialize $\boldsymbol{W}^\ell$, we sample an i.i.d. mixed-sign weight matrix $\tilde{\boldsymbol{W}}^\ell$ of half the size, then construct $\boldsymbol{W}^\ell$ according to

$$\boldsymbol{W}^1 = \begin{pmatrix} +\tilde{\boldsymbol{W}}^1 \\ -\tilde{\boldsymbol{W}}^1 \end{pmatrix} \qquad \boldsymbol{W}^\ell = \begin{pmatrix} \left[+\tilde{\boldsymbol{W}}^\ell\right]^+ & \left[-\tilde{\boldsymbol{W}}^\ell\right]^+ \\ \left[-\tilde{\boldsymbol{W}}^\ell\right]^+ & \left[+\tilde{\boldsymbol{W}}^\ell\right]^+ \end{pmatrix} \quad \ell > 1 \tag{16}$$

where $[\cdot]^+$ denotes positive rectification. These weights are mixed-sign for $\ell = 1$ and nonnegative for $\ell > 1$, consistent with the definition of VNNs. During training, the ON/OFF structure of the weights degrades, while the ON/OFF structure of the gating vectors is preserved.

The manner in which ON/OFF initialization avoids exploding forward propagation can be understood as follows. Let $(\lambda, \boldsymbol{v})$ denote an eigenvalue/vector pair of $\tilde{\boldsymbol{W}}^\ell$ and $(\lambda^+, \boldsymbol{v}^+)$ such a pair of abs$(\tilde{\boldsymbol{W}}^\ell)$. Then, $(\boldsymbol{v}, -\boldsymbol{v})/\sqrt{2}$ and $(\boldsymbol{v}^+, \boldsymbol{v}^+)/\sqrt{2}$ are eigenvectors of $\boldsymbol{W}^\ell$ with eigenvalues $\lambda$ and $\lambda^+$, respectively [34]. Thus, the spectrum of $\boldsymbol{W}^\ell$ is the union of the spectra of $\tilde{\boldsymbol{W}}^\ell$ and abs$(\tilde{\boldsymbol{W}}^\ell)$, which contains a large outlier eigenvalue from abs$(\tilde{\boldsymbol{W}}^\ell)$, a nonnegative i.i.d. matrix. However, each pre-activation vector has the form $\boldsymbol{h}^\ell = (\tilde{\boldsymbol{h}}^\ell, -\tilde{\boldsymbol{h}}^\ell)$, and the projection of $\boldsymbol{h}^\ell$ onto the eigenvector of the outlier eigenvalue is zero since $(\boldsymbol{v}^+, \boldsymbol{v}^+) \cdot (\tilde{\boldsymbol{h}}, -\tilde{\boldsymbol{h}}) = 0$.

In BP, the backpropagated signal lacks the ON/OFF structure of the forward-propagated signal and therefore possesses a component along the eigenvector of the outlier eigenvalue, resulting in a growing backward pass. Our experiments used an optimizer with a normalizing effect (namely, Adam), preventing large weight updates in early layers when using BP [35]. Importantly, when using GEVB, well-behaved forward propagation is sufficient for well-behaved weight updates.

## 6  Experimental results

Here, we show that GEVB performs well in practice. To disentangle the impact on network performance of vectorization and nonnegativity, we trained vectorized and conventional networks, with

Table 1: MNIST test errors (%). Train errors are shown in parentheses if greater than 0.005. Errors corresponding to GEVB in VNNs are shown in color. For each type of architecture, the smallest GEVB or DFA test error is **bold**.

| | Vectorized networks | | | |
| | Nonnegative | | Mixed-sign | |
| | GEVB | BP | GEVB | BP |
|---|---|---|---|---|
| Fully connected | **1.87 (0.06)** | 1.93 (0.05) | 2.32 (0.33) | 1.84 (0.07) |
| Convolutional | 2.33 (1.03) | 1.3 (0.19) | 1.83 (0.83) | 0.8 |
| Locally connected | 1.78 | 1.64 | 1.84 (0.05) | 1.44 |

| | Conventional networks | | | |
| | Nonnegative | | Mixed-sign | |
| | DFA | BP | DFA | BP |
|---|---|---|---|---|
| Fully connected | 2.2 (0.3) | 1.36 | 2.09 (0.19) | 1.29 |
| Convolutional | **1.56 (0.67)** | 0.71 | 1.64 (0.42) | 0.65 |
| Locally connected | 1.98 (0.32) | 1.21 | **1.48 (0.09)** | 1.07 |

Table 2: CIFAR-10 test errors (%). Conventions are the same as in Table 1

| | Vectorized networks | | | |
| | Nonnegative | | Mixed-sign | |
| | GEVB | BP | GEVB | BP |
|---|---|---|---|---|
| Fully connected | **47.62 (1.25)** | 47.03 (0.72) | 48.86 (2.02) | 45.98 (0.78) |
| Convolutional | **33.74 (28.83)** | 30.85 (16.29) | 38.43 (20.59) | 30.54 (1.71) |
| Locally connected | 41.08 (0.83) | 41.01 (0.34) | 40.11 (0.83) | 38.77 (0.36) |

| | Conventional networks | | | |
| | Nonnegative | | Mixed-sign | |
| | DFA | BP | DFA | BP |
|---|---|---|---|---|
| Fully connected | 48.69 (3.28) | 45.42 (0.96) | 49.54 (2.88) | 45.69 (0.73) |
| Convolutional | 54.18 (48.43) | 32.13 (0.23) | 44.07 (17.25) | 28.8 (0.15) |
| Locally connected | 41.18 (10.62) | 35.51 | **39.41 (2.94)** | 32.32 |

and without a nonnegativity constraint. Vectorized networks were trained using GEVB and BP, and conventional networks were trained using DFA and BP (see Appendix J for DFA details). The nonlinearity in conventional networks was the scalar case of the VNN nonlinearity with $t = \pm 1$. Mixed-sign networks were initialized using He initialization, and nonnegative networks were initialized using ON/OFF initialization with an underlying He initialization [36]. In conventional nonnegative networks, the DFA feedback matrices were nonnegative [18]. We trained fully connected, convolutional, and locally connected architectures. Locally connected networks have the same receptive field structure as convolutional networks, but lack weight sharing [17]. We trained models on MNIST [37] and CIFAR-10 [38], using wider and deeper networks for CIFAR-10. We used Adam for a fixed number of epochs (namely, 190), stopping early at zero training error. For each experiment, we performed five random initializations. Training lasted ~10 days using five NVIDIA GTX 1080 Ti GPUs. See Appendices G, H, and I for details.

Our experimental results are summarized in Tables 1 and 2 (see Appendix A for learning curves). We first examined the impact of vectorization and nonnegativity on model performance by considering errors under BP training. Across tasks and architectures, imposing vectorization or nonnegativity typically increased BP test and train errors, and imposing both increased the errors more than imposing either feature on its own. However, these increases in error were modest. One exception was locally connected networks trained on CIFAR-10, whose test error increased by 8.69% upon imposition of vectorization and nonnegativity. As the train error increased by only 0.34%, these features primarily diminished the generalization ability, rather than the capacity, of the model.

Next, we compared GEVB to BP in vectorized networks. In fully and locally connected VNNs, GEVB achieved test and train errors similar to those of BP (0.14% and 0.59% maximum discrepancies in test error on MNIST and CIFAR-10, respectively). In convolutional VNNs, we observed a small performance gap between GEVB and BP (1.33% and 2.89% discrepancies in test error on MNIST and CIFAR-10, respectively). One possible reason for this gap is that weight sharing in convolutional

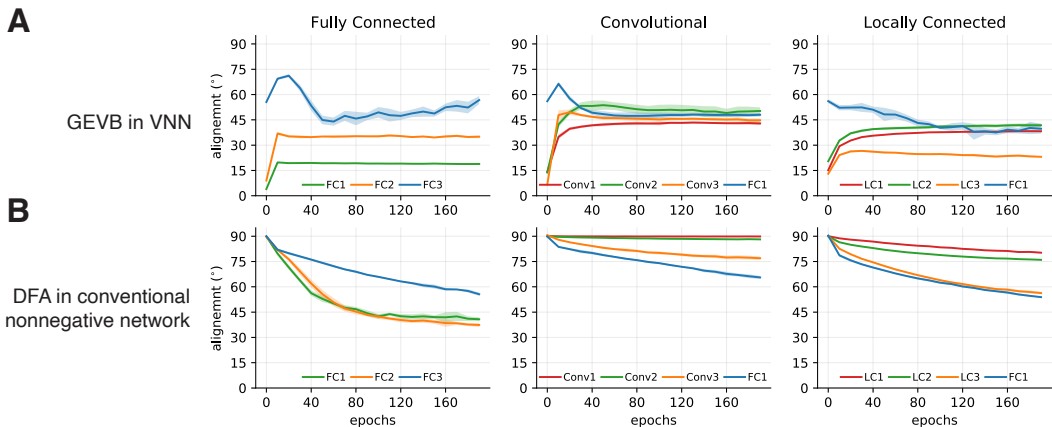

Figure 2: Gradient alignment angles over the course of training on CIFAR-10. In the legend, layers progress from early to late from left to right. Weight updates to the readout layer are equivalent to the gradient for GEVB and DFA, so these layers are not shown. **(A)** GEVB in VNNs. **(A)** DFA in conventional nonnegative networks. Error bars are standard deviations across five runs. See Appendix A for corresponding plots for mixed-sign networks and MNIST.

networks breaks GEVB's gradient sign match guarantee. Nevertheless, on CIFAR-10, GEVB in convolutional VNNs yielded substantially lower test error than all other convolutional experiments using GEVB or DFA, a feature we examine in detail below. Finally, in vectorized mixed-sign networks, the performance gap between GEVB and BP was larger than in VNNs across tasks and architectures. Rather than enjoying guaranteed gradient alignment by virtue of nonnegative-constrained weights, learning in these networks relied on the tendency of GEVB to generate a bias toward positive weights, a special case of the feedback alignment effect [13].

As indicated by the **bold** entries in Tables 1 and 2, GEVB in vectorized networks in some cases outperformed DFA in conventional networks. GEVB had lower test error than DFA for fully connected architectures on MNIST, and for fully connected and convolutional architectures on CIFAR-10. Moreover, GEVB tended to produce considerably lower train errors than DFA. This is particularly impressive in light of the fact that vectorized networks have higher errors than conventional networks under BP training. In cases where DFA outperformed GEVB, the gap in train error tended to be much smaller than the gap in test error, suggesting that VNNs were overfitting. This is plausible as VNNs have a factor of $K$ more parameters than conventional networks in the first hidden layer (Section 3.1).

To gain insight into our experimental results, we measured the alignment of both GEVB and DFA weight updates with the gradient over the course of training (Appendix E). Following previous works, we computed the alignment between single-example updates and averaged these values over each mini-batch [13, 19]. Consistent with the theoretical result of Section 4, GEVB alignment angles at initialization were around $\sim 45°$ for layer $L - 1$, and small (typically under $20°$) for layers $\ell < L - 1$ (Fig. 2A). Over the course of training, these small angles increased, plateauing around or below $45°$. By contrast, DFA exhibited alignment angles $\sim 90°$ at initialization, and these angles dropped over the course of training due to the feedback alignment effect (Fig. 2B). The plateaued alignment angles for GEVB were typically smaller than those for DFA. In convolutional networks, DFA updates failed to align with the gradient in convolutional layers, consistent with large errors as well as prior studies [16, 19, 20]. By contrast, GEVB had plateaued alignment angles around $45°$ for convolutional layers.

Finally, we examined the ability of GEVB to train convolutional layers by studying the learned representations on CIFAR-10 at the output of the convolutional layers. t-SNE embeddings of these representations before and after training revealed that GEVB, but not DFA, improved the level of class clustering (Fig. 3A; see Appendix K for t-SNE details) [39]. We quantified this improvement using a measure of cluster quality defined as $1 - (\frac{1}{K} \sum_{c=1}^{K} \overline{r_c^2}) / \overline{r^2}$, where $\overline{r_c^2}$ and $\overline{r^2}$ denote the average class-$c$ and average overall squared pairwise distance, respectively, in the t-SNE embedding. For all experiments, the cluster quality was slightly greater than zero at initialization and typically improved during training (Fig. 3B). DFA largely failed to improve cluster quality. By contrast, GEVB not only matched, but surpassed BP's improvement in cluster quality in vectorized networks.

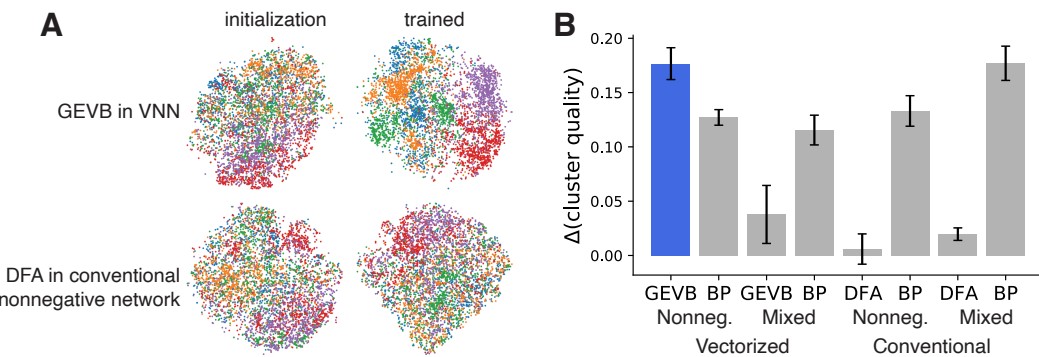

Figure 3: Comparison of learned representations at the output of convolutional layers in networks trained on CIFAR-10. **(A)** t-SNE embeddings of five image classes before and after training for GEVB in VNNs (top) and for DFA in conventional nonnegative networks (bottom). **(B)** Change in cluster quality (defined in main text) due to training for all eight convolutional experiments on CIFAR-10. GEVB in VNNs is shown in color. Error bars are standard deviations across five runs.

The observation that GEVB not only trains convolutional layers, but surpasses BP's improvement in cluster quality in vectorized networks, suggests that GEVB applies more pressure than BP on early layers to extract task-relevant features. Understanding how the representations generated by GEVB differ from those generated by BP is an avenue for future study. Such an understanding could enable disambiguation of cortical learning algorithms on the basis of neural recordings [40, 41].

## 7  Discussion

**Biological implementation:** Our work raises the question of how vector units and the GEVB learning rule could be implemented in neural circuits. Here, we suggest two possible implementations. For each solution, we describe the required circuit architecture, implementation of the VNN nonlinearity, and formulation of GEVB in terms of three-factor Hebbian plasticity [24, 25].

We first consider a solution in which each vector unit is implemented by a group of $K$ neurons. Consider a presynaptic group $j$ in layer $\ell - 1$ and a postsynaptic group $i$ in layer $\ell$. Neuron $\mu$ in group $j$ projects only to neuron $\mu$ in group $i$. Let $w_{ij\mu}^{\ell}$ denote the strength of this connection. As vectorized weight sharing is not enforced automatically in neural circuits, this weight has a $\mu$ index. The nonlinearity silences a group of neurons if the activity pattern of the group has positive alignment with the gating vector. This could be implemented through an inhibitory circuit mechanism. Given an input example and a target output, the network computes the error vector $\boldsymbol{e}$. The $K$ components of this error vector are then distributed throughout the network via $K$ distinct neuromodulatory signals, where the $\mu$-th signal is proportional to $e_\mu$. The $\mu$-th signal modulates only the $\mu$-th weight $w_{ij\mu}^{\ell}$. Then, each synapse undergoes a three-factor Hebbian update, $\Delta w_{ij\mu}^{\ell} = -e_\mu a_{j\mu}^{\ell-1} G(\boldsymbol{h}_i^{\ell})$. Here, $e_\mu$ is the global third factor, $a_{j\mu}^{\ell-1}$ is the presynaptic rate, and $G(\boldsymbol{h}_i^{\ell})$ is a function of the postsynaptic rate. We recover the GEVB learning rule by relaxing the strengths of the $K$ updated synapses between groups $j$ and $i$ to their average, $\frac{1}{K} \sum_\mu w_{ij\mu}^{\ell}$. Recent work demonstrates that lateral connections and anti-Hebbain plasticity can induce such a relaxation to the average during a sleep phase [42].

Next, we consider a solution in which vectorization unfolds in time. Each vector unit is implemented by a single neuron. In each of $K$ time bins, a different set of input neurons are active, producing a single component of the network output in each bin. As the network processes inputs at different times using the same weights, vectorized weight sharing is enforced automatically. As per the nonlinearity, each neuron must be active or inactive at all time bins according to the sign of $\boldsymbol{t} \cdot \boldsymbol{h}$, where $\boldsymbol{h}$ is the pre-activation vector, each element of which corresponds to a single time bin. Computing this inner product requires the full temporal input, violating causality. One workaround is to make the gating vector $\boldsymbol{t}$ sparse in all but its first component so that each neuron is active or inactive based on its input at the first time bin. Alternatively, gating could be implemented by an external inhibitory signal that is independent of the pre-activations [43–46]. In time bin $\mu$, each synapse $w_{ij}^{\ell}$ undergoes a three-factor Hebbian update, $\Delta w_{ij}^{\ell} = -e_\mu a_{j\mu}^{\ell-1} G(\boldsymbol{h}_i^{\ell})$. Here, $e_\mu$ is the global third factor at time bin $\mu$, $a_{j\mu}^{\ell-1}$ is

the presynaptic rate at time bin $\mu$, and $G(\boldsymbol{h}_i^\ell)$ is a function of the postsynaptic rate. Integrating these updates over all $K$ time bins performs a sum over $\mu$, implementing the GEVB rule.

These implementations make distinct predictions for the spatiotemporal structure of the neuromodulatory signal. The "grouped" implementation predicts that this signal is spatially heterogeneous, so that each synapse is modulated by the appropriate component of the error vector, and temporally uniform. By contrast, the "temporal" implementation predicts that this signal is temporally heterogeneous, so that the appropriate component of the error vector is broadcast during each time bin, and spatially uniform. Both implementations predict an important role for inhibition in silencing groups of neurons or individual neurons in a stimulus-dependent manner. Raman and O'Leary [26] have suggested that vectorized feedback signals could be a general mechanism for increasing the precision of credit assignment in neural circuits, providing evidence for this mechanism in the *Drosophila* connectome. Our findings are in line with this view.

**Prior uses of vector units:** Other works have used vector units for computational, rather than credit-assignment, purposes [47]. Capsule networks use vector-valued capsules to represent object parts [48] (see also Hinton [49]). Vector neuron networks (not to be confused with our proposed architecture, vectorized nonnegative networks) use three-dimensional vector units to achieve equivariance to the rotation group, SO(3), for three-dimensional point-cloud input data [50]. In gated linear networks, each unit stores a vector of parameters describing a probability distribution with the same support as the network output [43–45]. Gated linear networks use a form of vectorized weight sharing so that each unit represents a weighted geometric mixture of the distributions represented by units in the previous layer. In some cases, it is natural to constrain these weights to be nonnegative [44]. Vector units in VNNs can be interpreted as performing weighted geometric mixtures of categorical probability distributions, where each each vector activation encodes a vector of logits.

**Prior work on credit assignment:** In addition to works we have already mentioned, a wide variety of solutions to the credit assignment problem have been proposed. One approach involves dynamically generating approximate weight symmetry [23, 51, 52]. Another approach, target propagation, preserves information between layers by training a stack of autoencoders [17, 53]. In contrast to these methods, GEVB assumes a nonnegative feedforward pathway, obviating the need for a feedback pathway. Rather than minimizing an output loss, as in GEVB, several methods forego optimizing a global objective. For instance, some methods optimize neuron- or layer-local objectives [43, 54–56]. Notably, methods based on the information bottleneck [57] and contrastive prediction [58] enable training of DNNs using three-factor Hebbian learning. Several works have proposed local plasticity rules that perform unsupervised learning [59–62]. Still other approaches to credit assignment include equilibrium propagation [63, 64] and methods based on meta-learning [65–69]. Some works investigate the mechanism of credit assignment in specific biological structures [69, 70]. Exciting recent work shows that complex tasks can be learned in a segregated-dendrites model in which feedback connections trigger bursting events that induce synaptic plasticity [23]. In general, we are in need of a theoretical framework linking structure to learning capabilities in neural circuits [26].

**Scaling to high-dimensional outputs:** Using a large output dimension $K$ in VNNs is cumbersome as the computational complexity of the forward pass scales in $K$ (by contrast, the complexity of the backward pass does not scale in $K$). One solution is to use target vectors of size $K \sim \log C$ for $C$-way classification with $K$-bit binary, rather than one-hot, encodings of the target class index. This is especially appropriate if the binary encoding is congruent with hierarchical structure in the data. For example, such structure is present in ImageNet [71]. If vectorization unfolds in time, it is natural to produce more coarse-grained classifications of the input first. Also, note that the forward pass in VNNs is highly parallelizable: at each layer, the $K$ required matrix-vector products can be performed in parallel. Communication is then required to compute $\Theta(\boldsymbol{t} \cdot \boldsymbol{h})$ at each vector unit.

**Future directions:** Given GEVB's gradient sign match property and ability to train convolutional layers, our method is poised to scale to more complex tasks. GEVB often outperforms DFA, which successfully trains several modern architectures [19]. Biologically, the main work ahead lies in determining whether the core insights of our paper can be adapted to more realistic network models with spiking neurons and biophysically motivated plasticity rules [23]. Finally, the GEVB learning rule is designed to be aligned with the gradient, but gradient-based learning may not be optimal in scenarios such as few-shot or continual learning. Thus, while cortical circuits *could* plausibly use gradient-based learning, whether they *should* remains unclear. Future work should consider learning algorithms that are both biologically plausible and overcome shortcomings of BP [67, 72].

## Acknowledgements

We thank Jack Lindsey for insightful comments on an earlier version of this manuscript. We thank Greg Wayne, Jesse Livezey, and members of the Abbott lab for helpful pointers and discussion. Research was supported by NSF Neuronex 1707398 and the Gatsby Charitable Foundation GAT3708. The authors declare no competing financial interests.

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
