# Appendix for Credit Assignment Through Broadcasting a Global Error Vector

David G. Clark, L.F. Abbott, SueYeon Chung

## Contents

## A    Supplementary figures

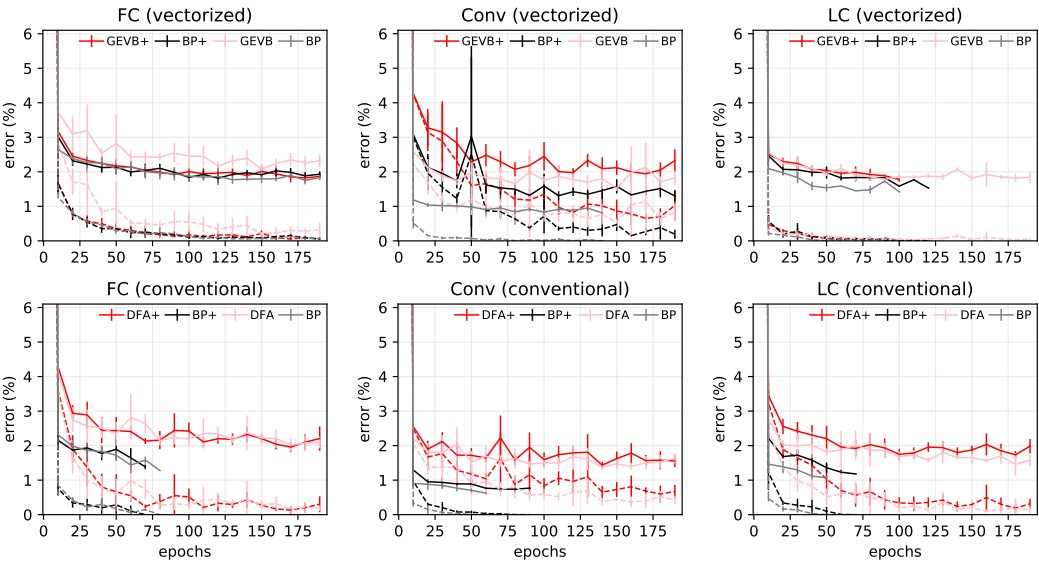

Figure 1: MNIST learning curves. Nonnegative-constrained networks have a "+" in the name of the learning rule. Thus, "GEVB+" corresponds to GEVB in VNNs. Solid line: test error. Dashed line: train error. Truncated curves reflect early stopping due to zero train error. Error bars are standard deviations across five runs.

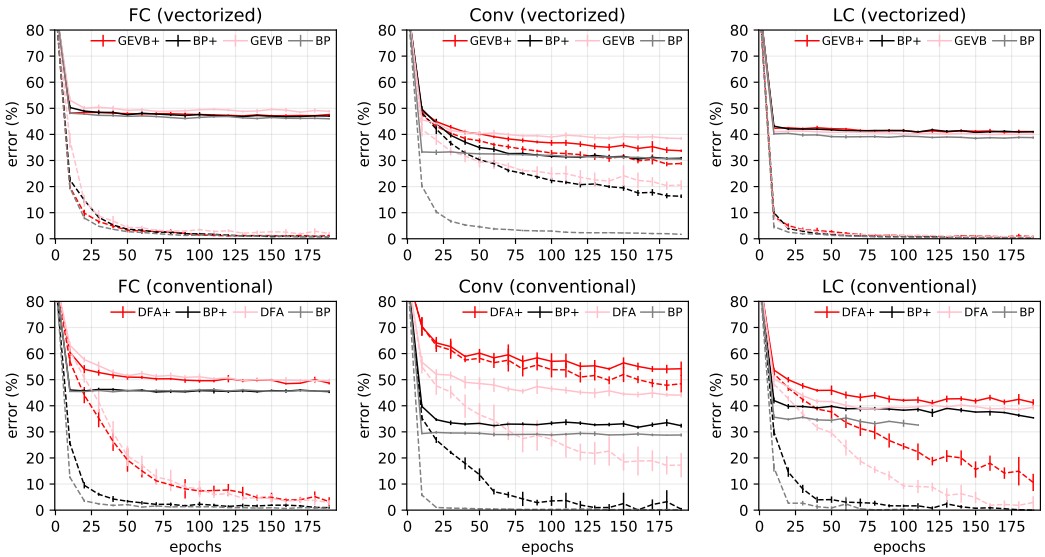

Figure 2: CIFAR-10 learning curves. Conventions are the same as in Fig. 1

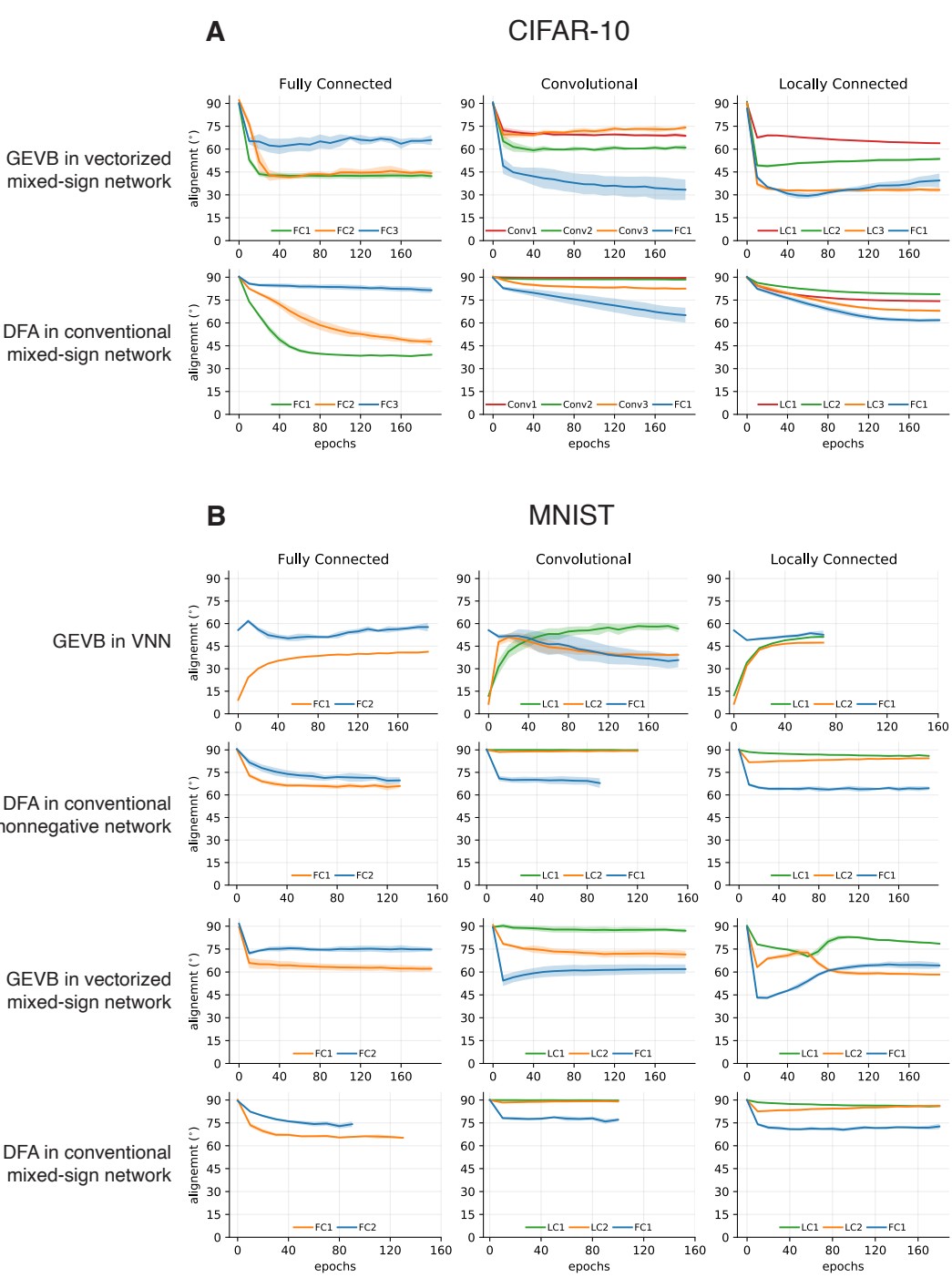

Figure 3: Alignment angles. (**A**) Mixed-sign networks trained on CIFAR-10. (**B**) Nonnegative and mixed-sign networks trained on MNIST. Truncated curves reflect early stopping due to zero train error. Conventions are the same as in Fig. 2 of the main text.

## B    Formulation of VNNs using vector input units

We can describe the input layer in a VNN as containing vector units, with vectorized weight-shared connections to the first hidden layer. In particular, given $n_0$ scalar input components $a_i^0$ ($i = 0, \ldots, n_0 - 1$), we can construct $Kn_0$ vector input units $a_{i\mu}^0$ ($i = 0, \ldots, Kn_0 - 1$) according to

$$a_{i\mu}^0 = \delta_{\mu\nu} a_j^0, \quad j = i \bmod n_0, \quad \nu = \left\lfloor \frac{i}{n_0} \right\rfloor \tag{A1}$$

where $\delta_{\mu\nu}$ is the Kronecker delta and $\lfloor \cdot \rfloor$ is the floor function (note that indices must start at zero for this formula to apply). This construction mimics the effect of having $n_0$ scalar inputs with all-to-all connectivity with components of the vector units in the first hidden layer.

## C    Assumption in GEVB sign match proof

Section 3.3 of the main text proves nonnegativity of $\hat{\delta}_i^\ell$. We require one assumption, stated here, to prove strict positivity of $\hat{\delta}_i^\ell$. Let a path refer to an inclusive sequence of units connecting two units in different layers of a VNN. Let the value of a path be the product of the weights along the path. We call a path active if all units along the path are in the active regimes of their nonlinearities. We borrow this terminology from [1]. To guarantee strict positivity of $\hat{\delta}_i^\ell$, we assume that, for all training examples, each hidden unit has at least one active path with nonzero value connecting it to the output unit.

When training VNNs, this assumption is violated for units in the last hidden layer that have zero weight onto the output unit, in which case the GEVB weight update is nonzero while the gradient is zero. This is insignificant in practice as the sign of the GEVB weight update is what the sign of the gradient *would* be if the weight were positive.

## D    Alternative derivation of GEVB

Starting from a general vectorized network, one can derive weight nonnegativity, the required form of the vector nonlinearity, and the GEVB learning rule itself from the requirement that the gradient sign is computable using only pre- and postsynaptic activity and the output error vector. Consider a vectorized network with arbitrary vector-to-vector nonlinearity $\Phi$. We then write down the derivative of the loss with respect to a particular weight $w_{ij}^\ell$ (i.e., Eq. 10 of the main text but leaving $\Phi$ arbitrary). Using the chain rule, we have

$$\frac{\partial \mathcal{L}}{\partial w_{ij}^\ell} = \sum_{\mu,\nu,\rho} e_\mu \frac{\partial a_\mu^L}{\partial a_{i\nu}^\ell} \Phi'_{\nu\rho}\left(\boldsymbol{h}_i^\ell\right) a_{j\rho}^{\ell-1} \tag{A2}$$

where $\Phi'_{\nu\rho}$ is the Jacobian of the vector nonlinearity. We then ask: under what conditions can the sign of this derivative be computed given only the presynaptic activation $a_{j\rho}^{\ell-1}$, the local Jacobian $\Phi'_{\nu\rho}\left(\boldsymbol{h}_i^\ell\right)$, and the error vector $e_\mu$? As per Eq. A2, this is possible if the global Jacobian, $\partial a_\mu^L / \partial a_{i\nu}^\ell$, is a positively scaled identity matrix. We therefore write down the backpropagation equation for the global Jacobian (i.e., Eq. 11 of the main text but leaving $\Phi$ arbitrary),

$$\frac{\partial a_\mu^L}{\partial a_{i\nu}^\ell} = \sum_{k=1}^{n_{\ell+1}} \sum_\rho \frac{\partial a_\mu^L}{\partial a_{k\rho}^{\ell+1}} \Phi'_{\rho\nu}\left(\boldsymbol{h}_k^{\ell+1}\right) w_{ki}^{\ell+1}. \tag{A3}$$

We see from this recurrence relation that requiring the global Jacobian to be a positively scaled identity matrix forces $w_{ki}^{\ell+1} \geq 0$ and forces $\Phi'_{\rho\nu}$ to be proportional to a nonnegatively scaled identity matrix. Having $\Phi'_{\rho\nu}$ proportional to a nonnegatively scaled identity matrix forces a nonlinearity of the form Eq. 7 of the main text. Setting the global Jacobian term in Eq. A2 to a positively scaled identity matrix yields the GEVB learning rule.

## E    Gradient alignment angle and relative standard deviation

Throughout this work, when computing the angular alignment of GEVB or DFA weight updates with the gradient, we do not include the derivative-of-nonlinearity term. Equivalently, these alignment

angles are computed with the assumption that all postsynaptic units are in the active regimes of their nonlinearities. This method follows the recommendations of Launay et al. [2] for measuring alignment angles in DFA. For a GEVB weight update, the alignment angle is given by

$$\cos\theta^\ell = \frac{\hat{\boldsymbol{\delta}}^\ell \cdot \mathbf{1}}{\|\hat{\boldsymbol{\delta}}^\ell\|\|\mathbf{1}\|}, \tag{A4}$$

where $\hat{\boldsymbol{\delta}}^\ell = \{\hat{\delta}_i^\ell\}_{i=1}^{n_\ell}$ comes from the gradient and the constant vector $\mathbf{1}$ comes from the GEVB weight update, which sets $\hat{\delta}_i^\ell = 1$. We define the empirical mean, variance, and relative standard deviation of the distribution of $\hat{\delta}_i^\ell$ in layer $\ell$ as

$$\mu_{\hat{\delta}^\ell} = \frac{1}{n_\ell}\sum_{i=1}^{n_\ell}\hat{\delta}_i^\ell, \quad \sigma_{\hat{\delta}^\ell}^2 = \frac{1}{n_\ell}\sum_{i=1}^{n_\ell}\left(\hat{\delta}_i^\ell - \mu_{\hat{\delta}^\ell}\right)^2, \quad r^\ell = \frac{\sigma_{\hat{\delta}^\ell}}{\mu_{\hat{\delta}^\ell}}. \tag{A5}$$

We have $\hat{\boldsymbol{\delta}}^\ell \cdot \mathbf{1} = n_\ell\mu_{\hat{\delta}^\ell}$, $\|\mathbf{1}\| = \sqrt{n_\ell}$, and

$$\|\hat{\boldsymbol{\delta}}^\ell\| = \sqrt{\sum_{i=1}^{n_\ell}\left(\hat{\delta}_i^\ell\right)^2} = \sqrt{n_\ell\left(\sigma_{\hat{\delta}^\ell}^2 + \mu_{\hat{\delta}^\ell}^2\right)}. \tag{A6}$$

Thus, Eq. A4 becomes

$$\cos\theta^\ell = \frac{1}{\sqrt{1 + \left(r^\ell\right)^2}}. \tag{A7}$$

Solving for $r^\ell$, we obtain

$$\left(r^\ell\right)^2 = \frac{1 - \cos^2\theta^\ell}{\cos^2\theta^\ell} = \frac{\sin^2\theta^\ell}{\cos^2\theta^\ell} = \tan^2\theta^\ell. \tag{A8}$$

For $r^\ell > 0$, we therefore have the simple relation $\tan\theta^\ell = r^\ell$.

## F  Concentration of relative standard deviation in wide networks

Here we prove the recurrence relation of Eq. 15 in the main text. Given the empirical mean and variance of the of the distribution of $\hat{\delta}_i^{\ell+1}$ in layer $\ell + 1$, we will compute the expectations of the empirical mean and variance of the distribution of $\delta_i^\ell$ in layer $\ell$ with respect to the randomness of the weights $\boldsymbol{W}^{\ell+1}$ and the gating variables $G_i^{\ell+1} = G(\boldsymbol{h}_i^{\ell+1})$. We assume that the layer-$(\ell + 1)$ weights are sampled i.i.d. from a distribution with mean $\mu_{w^{\ell+1}} > 0$ and variance $\sigma_{w^{\ell+1}}^2$. We assume that the gating variables $G_i^{\ell+1}$ are zero or one with equal probability.

Using Eq. 13 of the main text, the expected empirical mean is

$$\mathbf{E}\left[\mu_{\hat{\delta}^\ell}\right] = \mathbf{E}\left[\hat{\delta}_i^\ell\right] = \frac{n_{\ell+1}}{2}\mu_{w^{\ell+1}}\mu_{\hat{\delta}^{\ell+1}}. \tag{A9}$$

Meanwhile, the expected empirical variance can be written

$$\mathbf{E}\left[\sigma_{\hat{\delta}^\ell}^2\right] = \mathbf{E}\left[\left(\hat{\delta}_i^\ell - \frac{1}{n_\ell}\sum_j\hat{\delta}_j^\ell\right)^2\right] = \left(1 - \frac{1}{n_\ell}\right)\left(\mathbf{E}\left[\left(\hat{\delta}_i^\ell\right)^2\right] - \underset{i\neq j}{\boldsymbol{E}}\left[\hat{\delta}_i^\ell\hat{\delta}_j^\ell\right]\right). \tag{A10}$$

Toward evaluating $\mathbf{E}\left[\sigma_{\hat{\delta}_i^\ell}^2\right]$, we use Eq. 13 of the main text to compute

$$\mathbf{E}\left[\left(\hat{\delta}_i^\ell\right)^2\right] = \frac{1}{4}\mu_{w^{\ell+1}}^2\sum_{j\neq k}\hat{\delta}_j^{\ell+1}\hat{\delta}_k^{\ell+1} + \frac{n_{\ell+1}}{2}\left(\mu_{w^{\ell+1}}^2 + \sigma_{w^{\ell+1}}^2\right)\left(\mu_{\hat{\delta}^{\ell+1}}^2 + \sigma_{\hat{\delta}^{\ell+1}}^2\right) \tag{A11}$$

and

$$\underset{i\neq j}{\boldsymbol{E}}\left[\hat{\delta}_i^\ell\hat{\delta}_j^\ell\right] = \frac{1}{4}\mu_{w^{\ell+1}}^2\sum_{j\neq k}\hat{\delta}_j^{\ell+1}\hat{\delta}_k^{\ell+1} + \frac{n_{\ell+1}}{2}\mu_{w^{\ell+1}}^2\left(\mu_{\hat{\delta}^{\ell+1}}^2 + \sigma_{\hat{\delta}^{\ell+1}}^2\right). \tag{A12}$$

Substitution of Eqns. A11 and A12 into Eq. A10 yields

$$\mathbf{E}\left[\sigma_{\hat{\delta}^\ell}^2\right] = \left(1 - \frac{1}{n_\ell}\right)\frac{n_{\ell+1}}{2}\sigma_{w^{\ell+1}}^2\left(\mu_{\hat{\delta}^{\ell+1}}^2 + \sigma_{\hat{\delta}^{\ell+1}}^2\right)$$
$$\approx \frac{n_{\ell+1}}{2}\sigma_{w^{\ell+1}}^2\left(\mu_{\hat{\delta}^{\ell+1}}^2 + \sigma_{\hat{\delta}^{\ell+1}}^2\right). \tag{A13}$$

Altogether, we have

$$\frac{\sqrt{\mathbf{E}\left[\sigma_{\hat{\delta}^\ell}^2\right]}}{\mathbf{E}\left[\mu_{\hat{\delta}^\ell}\right]} = \sqrt{\frac{2}{n_{\ell+1}}}\frac{\sigma_{w^{\ell+1}}}{\mu_{w^{\ell+1}}}\sqrt{1 + (r^{\ell+1})^2}. \tag{A14}$$

We obtain the recurrence relation in Eq. 15 of the main text by approximating the empirical quantity $r^\ell$ as the ratio of expectations on the LHS of Eq. A14. This approximation is valid when the relative standard deviation of the weight distribution $\sigma_{w^{\ell+1}}/\mu_{w^{\ell+1}}$ is smaller than order $\sqrt{n_{\ell+1}}$, in which case the variance of $\mu_{\hat{\delta}^\ell}$ is small compared to its expectation.

## G  Architectures

Architectural details for MNIST and CIFAR-10 models are shown in Tables 1 and 2, respectively. We used the same architectures for vectorized and conventional networks. Note, however, that vectorized networks have a factor of $K$ more weight parameters in the first layer due to the lack of vectorized weight sharing in this layer. In convolutional networks, we used the same gating vector $t$ for all units in the same channel.

Table 1: MNIST architectures. **FC:** fully connected layer. **Conv:** convolutional layer. **LC:** locally connected layer. For fully connected layers, `layer_size` is shown. For convolutional and locally connected layers, (`num_channels`, `kernel_size`, `stride`, `padding`) are shown. The same architectures are used for conventional and vectorized networks.

| Fully connected | | Convolutional | | Locally connected | |
| --- | --- | --- | --- | --- | --- |
| FC1 | 1024 | Conv1 | $64, 3 \times 3, 1, 1$ | LC1 | $32, 3 \times 3, 1, 1$ |
| FC2 | 512 | AvgPool | $2 \times 2$ | AvgPool | $2 \times 2$ |
| | | Conv2 | $32, 3 \times 3, 1, 1$ | LC2 | $32, 3 \times 3, 1, 1$ |
| | | AvgPool | $2 \times 2$ | AvgPool | $2 \times 2$ |
| | | FC1 | 1024 | FC1 | 1024 |

Table 2: CIFAR-10 architectures. Conventions are the same as in Table 1.

| Fully connected | | Convolutional | | Locally connected | |
| --- | --- | --- | --- | --- | --- |
| FC1 | 1024 | Conv1 | $128, 5 \times 5, 1, 2$ | LC1 | $64, 5 \times 5, 1, 2$ |
| FC2 | 512 | AvgPool | $2 \times 2$ | AvgPool | $2 \times 2$ |
| FC3 | 512 | Conv2 | $64, 5 \times 5, 1, 2$ | LC2 | $32, 5 \times 5, 1, 2$ |
| | | AvgPool | $2 \times 2$ | AvgPool | $2 \times 2$ |
| | | Conv3 | $64, 2 \times 2, 2, 0$ | LC3 | $32, 2 \times 2, 2, 0$ |
| | | FC1 | 1024 | FC1 | 512 |

## H  Global error-vector broadcasting in convolutional networks

Convolutional networks use shared weights. When we apply GEVB in convolutional networks, we update each weight by the sum of all GEVB updates involving that weight. An equivalent description is the following. In a conventional convolutional network, weight updates are obtained by performing a convolution of the presynaptic activations with the postsynaptic backpropagated signal. When using GEVB, we replace the presynaptic signal with the inner product of the presynaptic vector activations and the output error vector; and the postsynaptic signal with the binary activation mask of the postsynaptic units.

## I   Training

Models were trained on an in-house GPU cluster. Running all 48 experiments in Tables 1 and 2 of the main text took $\sim$10 days using a single GPU, and we ran all experiments five times in parallel using five GPUs. We used the Adam optimizer with hyperparameters $\beta_1 = 0.9$, $\beta_2 = 0.999$, and $\epsilon = 10^{-8}$ [3]. We used a constant learning rate of $\alpha = 3 \times 10^{-4}$. Models were trained for 190 epochs or until the train error was zero at a checkpoint. Checkpoints were performed every 10 epochs. We used a mini-batch size of 128 for both datasets. We used the usual train/test splits for these datasets (60,000 and 50,000 training examples for MNIST and CIFAR-10, respectively; 10,000 test examples for each). In nonnegative networks, negative weights were set to zero following each update for layers $\ell > 1$.

## J   Direct feedback alignment

We used a PyTorch implementation of DFA from Launay et al. [4], modifying the code in the "TinyDFA" directory of their codebase.[1] To perform DFA in a multilayered network, this code samples one large random matrix, then uses submatrices of this matrix for the feedback to each layer. For mixed-sign networks, we sampled this matrix i.i.d. uniform over $[-1, 1]$. For nonnegative networks, we sampled this matrix i.i.d. uniform over $[0, 1]$, similar to Lechner [5].

## K   t-SNE

Before applying t-SNE, we projected the convolutional representations down to 600 dimensions using PCA. As the vectorized representations were higher dimensional than the conventional representations by a factor of $K = 10$, this projection put the dimensionalities on the same scale. We used a fast GPU implementation of t-SNE from Chan et al. [6]. We used the same hyperparameters as Launay et al. [4], namely, `perplexity` $= 20$, `learning_rate` $= 100$, and `n_iter` $= 5,000$.

## L   Potential negative societal impacts

The computational complexity of the forward pass in VNNs is larger by a factor of $K$ than in conventional networks. Thus, widespread adoption of VNNs in large-scale applications, however unlikely, could result in significant energy consumption.

## M   Summary of mathematical results

- Equivalence of scalar and vector VNN input formats (statement at the end of Section 3.1; proof in Appendix B)

- GEVB matches the sign of the gradient (statement and proof in Section 3.3; relies on a technical condition regarding "active paths" in Appendix C)

- Relationship between gradient alignment angle and relative standard deviation (statement in Section 4; proof in Appendix E)

- Exact agreement of GEVB updates with gradient in the infinite-width limit (statement in Section 4; proof in Appendix F)

- ON/OFF initialization results in a subnetwork with the same activations as a network of half the size with i.i.d. mixed-sign weights (statement in Section 5; proof is straightforward)

- Eigenvalues and eigenvectors of the ON/OFF matrices (statement in Section 5; proof is straightforward)

---

[1] https://github.com/lightonai/dfa-scales-to-modern-deep-learning