# OpenReview forum: "Credit Assignment Through Broadcasting a Global Error Vector"
_NeurIPS.cc/2021/Conference — NeurIPS 2021 Poster_

### Official Review · Reviewer_wVMT · 2021-07-12

**Rating:** 5
**Confidence:** 5

**Summary:**

This paper proposes to overcome an important biological implausibility of neural networks trained by backpropagation using two novelties: 1) to train neural networks using a new globally available error signal, 2) a new type of neural network with vector-valued units and nonnegative weights. Interestingly, the model shows that the proposed learning rules lead to update rules with the same sign as those obtained by gradient-descent.

This work justifies the biological plausibility of the learning rule by relating it to the growing literature on three-factor Hebbian rules and justifies the choice of nonnegative weights to observed cortical properties, as well as the presence of ON/OFF cells.

The performance of the proposed network is compared, on MNIST and CIFAR-10 datasets, against a very standard set network trained with BP, as well as more recent models trained using direct feedback alignment. The model outperforms competing DFA-trained models for fully connected and convolutional architectures but not for locally connected architectures.

**Limitations And Societal Impact:**

Two other important limitations of deep networks have not been addressed.

The first one is their lack of transparency, and this model remains "black-box," not offering a better understanding of how the learning is performed. The learning rule is compared to that of the gradient methods, which have shown to be hard to analyze and understand and do not address this problem. The claimed biological plausibility could have led to more understanding of the learning in the brain but fails to do so as it only vaguely addresses the topic. However, it appears to be a central point to the paper.

The second one is the bias that has been observed in many deep neural networks trained with backpropagation. Although it is not expected that the model would solve this problem, it still needs to be mentioned.

On a side note, the paper claims that a limitation of this work is the lack of experiments reported, i.e., only evaluated on MNIST and CIFAR-10. This paper is mostly experimental rather than theoretical; no theorem, proposition, or lemma is stated in the main text. As a result, more experiments are expected. This is thus not a limitation of the paper but a weakness of the submission.

**Main Review:**

Below is addressed the clarity.

This paper is clear and well-written, and the claims are clearly laid out. However, the notations are not easy to follow, in particular the not-standard choice of indexing of the network. It would have been useful to relate the notation with neural architecture presented in Figure 1, where it is very unclear how it relates to the rest of the notations used in the paper. It is also true for the "biological interpretation" (section 7) that does not easily relate to the architecture in Fig.1.
The results in Table 2 are not easy to compare as the competing models (bold) results are presented in different "sub"-tables limiting the readability. The rest of the results, Fig.2 and Fig.3, are clear and well described in the caption and the text.

Below is addressed the originality.

This work makes an interesting effort at providing "local" learning rules, which is an important limitation of BP; as such, the learning rule is novel. The comparison to three-factor Hebbian rules is mentioned but is not clearly analyzed. The work of [24], mentioned in the paper, offers a generic framework and discusses possible biological structures in which it could occur, unlike in this work. The comparison with this work needs to be clearly stated as the model claims to be biologically sensible due to the presence of the aforementioned third factor.
This work claims to be both theoretical and original; however, proposing an ad-hoc learning rule can hardly be qualified to be theoretical when its analysis lacks theorems, or propositions, or even lemmas. It is suggested to the author to either rewrite the paper more mathematically or to provide more experiments and biological realism. The next point relates to the lack of experiments and even mention of competing models that are potentially more biologically plausible than those presented in the paper. Indeed, pieces of work from [i] and [ii] could rely on standard unsupervised Hebbian/Hopfield models followed by an SVM (that can be easily trained with biologically plausible models) and reach competing performance on CIFAR-10. As mentioned in the limitation review, the lack of proper theoretical analysis, and biological implementation, of the algorithm does not allow for a weak experimental section. We invite the reader to strengthen at least two out of three of those contributions. Various other models have also been proposed [iii-v] that have been omitted to be cited, using local biologically plausible learning rules and using 3-factor Hebbian rules.

Below is addressed the significance.

This work is definitely a step in the right direction in building biologically plausible algorithms but falls short of making a significant contribution. As mentioned above, the paper needs substantial improvement in at least one of the following contributions, theoretical, experimental, or biological realism.


[i] Krotov, Dmitry, and John J. Hopfield. "Unsupervised learning by competing hidden units." Proceedings of the National Academy of Sciences 116.16 (2019): 7723-7731.
[ii] Bahroun, Yanis, and Andrea Soltoggio. "Online representation learning with single and multi-layer Hebbian networks for image classification." International Conference on Artificial Neural Networks. Springer, Cham, 2017.

[iii] Pogodin, Roman, and Peter E. Latham. "Kernelized information bottleneck leads to biologically plausible 3-factor Hebbian learning in deep networks." NeurIPS 2020/
[iv] Golkar, Siavash, et al. "A biologically plausible neural network for local supervision in cortical microcircuits." NeurIPS 2020.
[v] Qin, Shanshan, Nayantara Mudur, and Cengiz Pehlevan. "Contrastive Similarity Matching for Supervised Learning." Neural Computation 33.5 (2021): 1300-1328.

**Time Spent Reviewing:**

4

---

> ### Author Response · Authors · 2021-08-10
> **Response to reviewer wVMT**
>
> We thank reviewer wVMT for their thoughtful review of our paper. The reviewer highlighted three areas in which they believed our paper required improvement, namely, mathematical rigor, biological plausibility, and breadth of comparisons to other works. The reviewer requested that we enhance the paper in a subset of these areas. In the revised paper, we will clarify that our work is in fact rooted in several mathematical derivations (see below). Moreover, the revised paper will discuss biological implementations of our proposed learning rule with a high degree of detail (see collective response). Thus, we believe we have satisfied reviewer wVMT's request.
>
> - *"This work claims to be both theoretical and original; however, proposing an ad-hoc learning rule can hardly be qualified to be theoretical when its analysis lacks theorems, or propositions, or even lemmas. It is suggested to the author to...rewrite the paper more mathematically..." The reviewer later states: "This paper is mostly experimental rather than theoretical; no theorem, proposition, or lemma is stated in the main text."*
>
> 	Our paper contains a number of theoretical claims with proofs. We simply elected to include these results in the text rather than label them as theorems, propositions, or lemmas. In the revised paper, we will clarify that the following theoretical results constitute concrete mathematical derivations.
>
> 	- Proposition 1: Equivalence of scalar and vector VNN input formats (statement in lines 123--126; proof in Appendix B).
> 	- Theorem 1: GEVB matches the sign of the gradient (statement and proof in Section 3.3; relies on a technical condition regarding "active paths" in Appendix C).
> 	- Proposition 2: Relationship between gradient alignment angle and relative standard deviation (statement in lines 156--164; proof in Appendix D).
> 	- Theorem 2: Exact agreement of GEVB updates with gradient in the infinite-width limit (statement in Section 4; proof in Appendix E; relies on Proposition 2).
> 	- Theorem 3: ON/OFF initialization results in a subnetwork with the same activations as a network of half the size with i.i.d. mixed-sign weights (statement in Section 5; proof details will be included in the Appendix of the revised paper).
> 	- Proposition 3: Eigenvalues and eigenvectors of the ON/OFF matrices (statement in lines 199--204; proof details will be included in the Appendix of the revised paper).
>
> 	Finally, note that the nonnegativity of VNNs and the GEVB learning rule can be "derived" in the following way. Consider a vectorized network with arbitrary vector-to-vector nonlinearity $\Phi$. We then write down the derivative of the loss with respect to a particular weight $w^{\ell}\_{ij}$ (i.e., Eq. 10 but leaving $\Phi$ arbitrary). Using the chain rule, we have
> 	$$\frac{\partial \mathcal{L}}{\partial w^{\ell}\_{ij}} = \sum\_{\mu, \nu, \rho} e\_{\mu} \frac{\partial a^L\_{\mu}}{\partial a^{\ell}\_{i \nu}} \Phi'\_{\nu \rho}\left( \mathbf{h}^{\ell}\_i \right) a^{\ell-1}\_{j \rho}$$
> 	where $\Phi'\_{\nu \rho}$ is the Jacobian of the vector nonlinearity. We then ask: under what conditions can the sign of this derivative be computed given only the pre-activation $a^{\ell-1}\_{j \rho}$, the local Jacobian $\Phi'\_{\nu \rho}\left( \mathbf{h}^{\ell}\_i \right)$, and the global error vector $e\_{\mu}$? This is possible if the global Jacobian $\frac{\partial a^L\_{\mu}}{\partial a^{\ell}\_{i \nu}}$ is equal to a positively scaled identity matrix. This, in turn, forces nonnegative weights and a nonlinearity of the form Eq. 7. We will include this derivation, with more detail, in the revised paper.
>
> - *"The comparison to three-factor Hebbian rules is mentioned but is not clearly analyzed. The work of [24], mentioned in the paper, offers a generic framework and discusses possible biological structures in which it could occur, unlike in this work. The comparison with this work needs to be clearly stated as the model claims to be biologically sensible due to the presence of the aforementioned third factor."*
>
> 	We will explicitly describe biological structures in which VNNs could be implemented, as well as the relationship between the GEVB learning rule and three-factor Hebbian learning (see collective response).
>
>
> - *"The next point relates to the lack of experiments and even mention of competing models that are potentially more biologically plausible than those presented in the paper. Indeed, pieces of work from [i] and [ii] could rely on standard unsupervised Hebbian/Hopfield models followed by an SVM (that can be easily trained with biologically plausible models) and reach competing performance on CIFAR-10."*
>
> 	We agree that our paper would benefit from a more complete review of biologically plausible learning algorithms (see also our response to reviewer oN9Y). We note that [i] performs very poorly on CIFAR-10, likely because only a single fully connected hidden layer is used. Similarly, while the method of [ii] shows impressive results on CIFAR-10 in a large convolutional network, this method falls far behind BP on a smaller fully connected network. By contrast, GEVB consistently matches the performance of BP in fully connected networks. Overall, [i] and [ii] exemplify a fundamentally different approach to credit assignment than the one we have pursued in which learning in hidden layers is not based on following the gradient of a global loss function. While such approaches to solving the credit assignment problem are important, we believe that comparisons to more directly related methods (i.e., BP, DFA, and sign-constrained DFA) are sufficient for the scope of a NeurIPS paper. Moreover, our proof that GEVB matches the sign of the gradient of a global loss function provides a theoretical basis for predicting how GEVB should perform relative to other methods.
>
> - *"...the notations are not easy to follow, in particular the not-standard choice of indexing of the network."*
>
> 	As no standard notation for vector units exists, we did our best to choose a sensible convention. We will clarify our notations in Section 3.1 of the revised paper.
>
> - *"It would have been useful to relate the notation with neural architecture presented in Figure 1, where it is very unclear how it relates to the rest of the notations used in the paper."*
>
> 	In the architectural diagram of VNNs (Fig. 1D), each arrow represents a vector unit. In Fig. 1 A--D, circles represent conventional scalar units. We will include this information in the figure caption.
>
> - *"It is also true for the "biological interpretation" (section 7) that does not easily relate to the architecture in Fig.1."*
>
> 	The revised paper will include a figure containing visual representations of the proposed biological implementations of VNNs (see collective response).
>
> - *"The results in Table 2 are not easy to compare as the competing models (bold) results are presented in different "sub"-tables limiting the readability."*
>
> 	We will improve the readability of our tables in the revised paper.

---

> > ### Comment · Reviewer_wVMT · 2021-08-17
> > **Response**
> >
> > I appreciate the effort made by the authors to respond to the reviews, which certainly strengthens the paper. I nonetheless believe that this work is still slightly below the acceptance threshold for a venue like NeurIPS. I will update my score accordingly.

---

### Official Review · Reviewer_Pci4 · 2021-07-16

**Rating:** 6
**Confidence:** 4

**Summary:**

This paper proposes (i) a learning rule called global error-vector broadcasting (GEVB), and (ii) a class of DNNs called
vectorized nonnegative networks (VNNs) in which this rule operates - to solve the credit assignment problem. While Backpropgation (BP) is remarkably effective at training deep neural networks, it is biologically implausible because of the weight transport problem. In this paper, the weight transport problem is circumvented by globally broadcasting the error signal to all hidden units in the network. However, in order to accommodate the vector error signal a new network model with vector units is proposed.

Further, by deriving gradients using a BP like approach, the authors demonstrate gradient alignment with the GEVB learning rule. The performance of VNNs trained with GEVB is evaluated on MNIST and CIFAR-10 and compared with BP on VNNs and conventional networks.



**Limitations And Societal Impact:**

The limitations of the work have been discussed by the authors - performance discrepancies on more challenging tasks, biological plausibility, and the computational complexity of the forward pass in VNNs.

There are no other potential negative societal impacts or ethical concerns.


**Main Review:**

__Significance__: Understanding credit assignment in biological networks is an important problem. Several approaches have been considered to circumvent the weight transport problem of BP, such as node perturbation and feedback alignment. In this work, an interesting alternative approach based on broadcasting an error signal is considered. However, the impact of this approach hinges on finding a biological plausible implementation of VNNs.

__Originality__:  The main idea in this paper is the broadcasting of the error signal to all hidden units of the network. The broadcast error is combined in a Hebbian-like learning rule for each weight. This is in contrast to sending a unique projection of the error signal to each hidden unit as in direct feedback alignment. However, in order to accommodate a vector error signal, the units in the network are modified to have vector activities instead of scalars. This work also provides analyses that demonstrate gradient matching for the GEVB rule.

__Quality__: The experimental results show the impact of vectorization and nonnegativity of weights on the VNN model performance. In most cases the performance of VNNs trained with GEVB was quite close to BP. Gradient alignment analysis reveals that GEVB results in better alignment than DFA on average. While the model itself is interesting, the biological implementation of vectorization is not quite convincing.  Another important issue is that training large networks with this approach is computationally challenging.

__Clarity__:  The paper is well presented and easy to follow.
Minor: Caption for figure 2 has label (A) for both panels.

__Questions__:
* An important consideration is how the error signal is broadcast to all units in the network. Do we expect the influence of a neuromodulatory system to be exactly the same on all units in a large network? How would some stochasticity in the error signal received at each unit affect the performance?
* The activation function used here is of a very specific form. How could this be biologically implemented?
* Non-negativity is important for gradient sign matching. And this corresponds well with the fact that a majority of cortical neurons are excitatory. However, inhibitory neurons do constitute a small but crucial neuronal class in the cortex. How well would VNNs with GEVB work without the sign constraint on the weights?
* The comparison of learned representations was done using networks trained just on CIFAR-10. This is an interesting analysis but is it conclusive enough to say GEVB in VNNs learns better representations?



**Time Spent Reviewing:**

3-4

---

> ### Author Response · Authors · 2021-08-10
> **Response to reviewer Pci4**
>
> We thank reviewer Pci4 for their thorough review of our work, which they state presents "an interesting alternative approach" to solving the credit assignment problem. In what follows, we believe we have addressed all questions and concerns.
>
> - *"While the model itself is interesting, the biological implementation of vectorization is not quite convincing."*
>
> 	Please see the expanded discussion regarding biological implementations in our collective response.
>
> - *"Another important issue is that training large networks with this approach is computationally challenging."*
>
> 	The forward pass in VNNs is computationally expensive, but highly parallelizable. In particular, at each layer, the $K$ required matrix-vector products can be performed in parallel. Communication is then required to compute the inner products $\mathbf{t} \cdot \mathbf{h}$ for the nonlinearities. We will include this potential optimization in Section 7 of the revised paper.
>
> - *"An important consideration is how the error signal is broadcast to all units in the network. Do we expect the influence of a neuromodulatory system to be exactly the same on all units in a large network? How would some stochasticity in the error signal received at each unit affect the performance?"*
>
> 	We agree that stochasticity in the feedback signal is an important consideration. The GEVB rule handles a moderate amount of noise gracefully. In particular, as long as the noise is not strong enough to change the signs of a large fraction of inner products $\mathbf{a} \cdot \mathbf{e}$, where $\mathbf{a}$ is a post-activation vector and $\mathbf{e}$ is the global error vector, weight updates remain correlated with the gradient.
>
> - *"The activation function used here is of a very specific form. How could this be biologically implemented?"*
>
> 	We will address the nonlinearity in our expanded discussion of biological implementations (see collective response).
>
> - *"Non-negativity is important for gradient sign matching. And this corresponds well with the fact that a majority of cortical neurons are excitatory. However, inhibitory neurons do constitute a small but crucial neuronal class in the cortex. How well would VNNs with GEVB work without the sign constraint on the weights?"*
>
> 	We performed this experiment and included in Section 6 of the paper (lines 239--243; Tables 1, 2; Fig. 3B; Supplementary Figs. 1--3). As we described, learning occurs in mixed-sign networks because the GEVB rule induces a bias toward positive weights, a special case of the feedback alignment effect. Finally, note that the nonlinearity relies on inhibition (see collective response).
>
> - *"The comparison of learned representations was done using networks trained just on CIFAR-10. This is an interesting analysis but is it conclusive enough to say GEVB in VNNs learns better representations?"*
>
> 	We agree that this experiment does not demonstrate that GEVB learns better representations than other learning rules (such as DFA) in general, nor is it clear what constitutes a "better representation." Rather, our conclusions are that 1) GEVB, unlike DFA, generates useful representations in hidden layers of convolutional networks and 2) in the particular task we studied, GEVB seems to apply more pressure than BP on early layers to learn useful representations A more thorough investigation of representations generated by GEVB is an avenue for future study.

---

> > ### Comment · Reviewer_Pci4 · 2021-09-01
> > **Response**
> >
> > Thank you for the detailed responses to my questions. Most of the concerns have been addressed. The discussion regarding the biological plausibility in the collective response is intriguing, and there are several aspects of the proposed models worthy of future exploration.

---

### Official Review · Reviewer_oN9Y · 2021-07-16

**Rating:** 8
**Confidence:** 4

**Summary:**

The paper proposes to circumvent the need of backpropagation in deep networks by introducing two architectural changes. First, the weights are restricted to be non-negative so the chain rule simplifies significantly. Second, each unit in a network is represented by a vector to match the dimensionality of the error signal in the top layer, further simplifying the chain rule. The resulting weight update approximates backpropagation similar to the sign symmetry method.

**Limitations And Societal Impact:**

The authors have adequately addressed the limitations and potential negative societal impact of their work.

**Main Review:**

The paper proposes a very novel and intriguing way to approximate backprop. The discussion on feedback alignment-type (FA) algorithms is missing two important papers, however. First, sign symmetry [1] is another FA-type algorithm that uses signs of the weights for the backward pass, and I think this algorithm works for a similar reason (although GEVB should be a tighter approximation due to non-negative weights; [26] in the paper seems relevant too, but [1] was published a few years earlier). Also, [2] shows how to introduce plasticity in the backward connections of FA to bring it on par with backprop, so it’s worth including into the discussion. Finally, non-FA-type methods [3-7] might be worth discussing (and comparing in experiments) too.

Quality: The submission is technically sound, and provides sufficient theoretical and experimental justification.

Clarity: The paper is well-written. One comment: line 236: “One possible reason for this gap is that weight sharing in convolutional networks breaks GEVB’s gradient sign match guarantee.” Does it break it though? Convolutions would add a sum over $i$ in Eq.14, resulting in $(\\sum_i \\delta_i G(h_i)) (\\sum_\\mu ...)$. GEVB should change $\\sum_i \\delta_i G(h_i)$  to $\\sum_i G(h_i)$, which doesn’t seem to break the approximation due to non-negativity of $G$.

Significance: Overall, the paper is a significant result with future potential. The major limitation of the paper is weight sharing within vector-valued units. While the authors do provide a potential mechanism for that (through time unfolding), it implies a rather complicated training curriculum (especially for large K). However, this limitation is unconventional and might be solved with future methods/experimental evidence.

[1] How Important is Weight Symmetry in Backpropagation?, Qianli Liao, Joel Z. Leibo, Tomaso Poggio

[2] Deep Learning without Weight Transport, Mohamed Akrout, Collin Wilson, Peter C. Humphreys, Timothy Lillicrap, Douglas Tweed

[3] Deep supervised learning using local errors, Hesham Mostafa, Vishwajith Ramesh, and Gert Cauwenberghs

[4] Training neural networks with local error signals, Arild Nøkland and Lars Hiller Eidnes

[5] Kernelized information bottleneck leads to biologically plausible 3-factor Hebbian learning in deep networks, Roman Pogodin and Peter Latham

[6] Local plasticity rules can learn deep representations using self-supervised contrastive predictions, Bernd Illing, Jean Ventura, Guillaume Bellec, Wulfram Gerstner

[7] Scaling Equilibrium Propagation to Deep ConvNets by Drastically Reducing Its Gradient Estimator Bias, Axel Laborieux, Maxence Ernoult, Benjamin Scellier, Yoshua Bengio, Julie Grollier and Damien Querlioz


**Time Spent Reviewing:**

3

---

> ### Author Response · Authors · 2021-08-10
> **Response to reviewer oN9Y**
>
> We are pleased that reviewer oN9Y has deemed our work "very novel and intriguing." We thank the reviewer for their thougthful comments and for pointing out several relevant references. In what follows, we believe we have addressed all questions and concerns.
>
> - *"The discussion on feedback alignment-type (FA) algorithms is missing two important papers, however. First, sign symmetry [1] is another FA-type algorithm that uses signs of the weights for the backward pass, and I think this algorithm works for a similar reason (although GEVB should be a tighter approximation due to non-negative weights; [26] in the paper seems relevant too, but [1] was published a few years earlier). Also, [2] shows how to introduce plasticity in the backward connections of FA to bring it on par with backprop, so it's worth including into the discussion."*
>
> 	We agree that the paper would benefit from an expanded discussion of related methods. We will include three references to the sign symmetry algorithm in Section 1 of the revised paper (Liao et al., AAAI 2016; Xiao et al., ICLR 2018; Moskovitz et al., arXiv:1812.06488 2018). Sign symmetry is a special case of feedback alignment in which the feedback matrices have the same sign as the feedforward matrices. In contrast, GEVB has no feedback parameters and generates weight updates with the same sign as the gradient. As the reviewer points out, sign symmetry has no such guarantee and therefore results in a looser approximation of the gradient. We will also include a reference to Akrout et al. (NeurIPS 2019) in Section 1. Note that both sign symmetry and the method of Akrout et al. solve the credit assignment problem by enforcing that the feedback pathway is aligned with the feedforward pathway. By contrast, GEVB enforces that the feedforward pathway is nonnegative, removing the need for a feedback pathway.
>
> - *"Finally, non-FA-type methods [3-7] might be worth discussing (and comparing in experiments) too."*
>
> 	We agree and will include these methods in Section 1 of the revised paper.
>
> - *"One comment: line 236: "One possible reason for this gap is that weight sharing in convolutional networks breaks GEVB's gradient sign match guarantee." Does it break it though? Convolutions would add a sum over i in Eq.14...which doesn't seem to break the approximation due to non-negativity of G"*
>
> 	Convolutions involve a sum over $j$ (the presynaptic index) in addition to $i$ (the postsynaptic index). This sum over $j$ breaks the sign-match guarantee. Nevertheless, GEVB was highly effective in training convolutional networks. We will clarify this in our discussion of applying GEVB in convolutional networks (Appendix G).

---

> > ### Comment · Reviewer_oN9Y · 2021-08-17
> > **Comment**
> >
> > Thank you for the clarifications given here and in the other responses! I think the concerns raised in the reviews are mostly addressed, and I'm happy to leave my current score (8) as it is.
> >
> > > Convolutions involve a sum over (the presynaptic index) in addition to (the postsynaptic index)
> >
> > I guess the weight update selects $j$ as a function of $i$ rather than sums over $j$; but I now realise where I was wrong.

---

### Official Review · Reviewer_osc2 · 2021-07-18

**Rating:** 6
**Confidence:** 4

**Summary:**

This paper presents a biologically plausible learning method, called global error-vector broadcasting (GEVB). The authors also propose vectorized nonnegative networks (VNNs) on which the GEVB method operates. The key property of the GEVB method is that the gradient has the same sign as the ground truth gradient returned by back-propagation (BP) on VNNs. Experiments on MNIST and CIFAR are conducted to show that the proposed method can sometimes match the performance of BP and sometimes outperform competitors like direct feedback alignment.


**Ethical Concerns:**

None.

**Limitations And Societal Impact:**

The authors have discussed the computational cost as the potential negative societal impact, which I think is fair.

**Main Review:**

Pros:

1, The proposed GEVB essentially propagates the error signal from the readout layer to all other layers in a direct manner. It seems to be more efficient and biologically more plausible compared to BP since it avoids the symmetric weights and the successive computation of gradients across layers. I think this is a novel and interesting contribution towards more biologically plausible learning methods.

2, The authors also show that in the limit of infinite network width, the proposed update in GEVB is proportional to the exact gradient at the initialization. This observation provides some potentially interesting connections to the recent study on infinite wide networks like NTK.

3, The empirical results show that the proposed learning method 1) does converge and 2) is comparable or even outperforms other alternatives. In particular, it successfully trains convolutional layers whereas other biologically plausible methods like direct feedback alignment (DFA) have some troubles.

Cons and Questions:

1, I agree that the proposed GEVB is more biologically plausible compared to BP. However, the introduction of this new type of architecture, i.e., VNNs, seems to be unsatisfying. In particular, do you have some evidence to support that VNNs are biologically more plausible than other common architectures?

2, Why do you need the random gating vectors per layer? If I understood correctly, your method and theory still work if those random gating vectors are removed. It is unclear to me what the motivation and the impact of this additional design choice are. More importantly, this random gating would inevitably bring some stochasticity to the final predictions. Therefore, it would be important to do a thorough ablation study on this component and show the standard deviations of the final performances corresponding to the aforementioned stochasticity.

3, Could you explain why the training and testing errors of GEVB are inconsistent for fully-connected and convolutional VNNs, i.e., better than or comparable to BP for fully-connected VNNs and significantly worse than BP for convolutional VNNs? Moreover, for convolutional VNNs, why does GEVB outperform DFA on CIFAR but perform worse on MNIST?

4, When you compute the gradient alignment angle, do you compute the angle per sample and then take the average, or do you compute the angle between gradients averaged over mini-batch? I am curious whether these two have significant differences since matching the signs of individual gradients is different from matching the signs of gradients averaged over mini-batch. From the perspective of optimization, we should care more about the latter.

5, In Fig. 2, could you comment on why the angle curve of the last readout (FC) layer seems to be diverging on fully connected VNNs? Also, why do the curves of the last readout (FC) layer first increase and then decrease rather than directly decrease on both fully-connected and convolutional VNNs?

6, The current method seems to be surprisingly slow, e.g., taking 10 days of GPU training even on CIFAR and MNIST. Could you comment on what are the exact factors that slow down the training?

## After Rebuttal

I'd like to thank the authors for the detailed response! It does address most of my concerns.
I was expecting authors to comment in detail on if there is evidence in neural science to back up the existence of some form of the VNNs.
However, I found the current response on this point less informative.
In addition, I agree with *Reviewer wVMT* that thorough experimental comparison on more datasets would significantly improve the paper.
In summary, I would like to keep my original rating.


**Time Spent Reviewing:**

4

---

> ### Author Response · Authors · 2021-08-10
> **Response to reviewer osc2**
>
> We thank reviewer osc2 for their careful review of our paper and for finding our work "novel and interesting." In what follows, we believe we have addressed all questions and concerns.
>
> - *"I agree that the proposed GEVB is more biologically plausible compared to BP. However, the introduction of this new type of architecture, i.e., VNNs, seems to be unsatisfying. In particular, do you have some evidence to support that VNNs are biologically more plausible than other common architectures?"*
>
> 	Cortical projection neurons are excitatory. The role for excitatory connections in simplifying credit assignment in multilayered networks has not been previously observed to our knowledge. We believe that this represents a step forward in understanding how the credit assignment problem could be solved in known biological structures. In the revised paper, we will include a substantially expanded discussion of biological implementations of VNNs (see collective response). Crucially, each implementation makes testable predictions regarding the structure of the neuromodulatory signal.
>
> - *"Why do you need the random gating vectors per layer? If I understood correctly, your method and theory still work if those random gating vectors are removed. It is unclear to me what the motivation and the impact of this additional design choice are. More importantly, this random gating would inevitably bring some stochasticity to the final predictions. Therefore, it would be important to do a thorough ablation study on this component and show the standard deviations of the final performances corresponding to the aforementioned stochasticity."*
>
> 	The gating vectors are sampled at initialization and held constant throughout training and inference. Thus, they do not inject noise into the final predictions (rather, they are a form of "quenched disorder"). We will clarify this in Section 3.1 of the revised paper.
>
> - *"Could you explain why the training and testing errors of GEVB are inconsistent for fully-connected and convolutional VNNs, i.e., better than or comparable to BP for fully-connected VNNs and significantly worse than BP for convolutional VNNs? Moreover, for convolutional VNNs, why does GEVB outperform DFA on CIFAR but perform worse on MNIST?"*
>
> 	Weight sharing in convolutional networks breaks the sign-match guarantee of GEVB (lines 236--237; see also our response to reviewer oN9Y). This could contribute to the larger gap between GEVB and BP in convolutional networks compared to fully connected networks, where the sign-match guarantee holds. As for MNIST, note that the gap in train error between GEVB in a VNN and DFA in a conventional nonnegative network is much smaller than the gap in test error, suggesting that the VNN could be overfitting. This is plausible as 1) the VNN contains a factor of $K$ more parameters in the first hidden layer (lines 109–111) and 2) MNIST is a simple task. We will expand the discussion of our experimental results, including both of these observations, in Section 6 of the revised paper.
>
> - *"When you compute the gradient alignment angle, do you compute the angle per sample and then take the average, or do you compute the angle between gradients averaged over mini-batch? I am curious whether these two have significant differences since matching the signs of individual gradients is different from matching the signs of gradients averaged over mini-batch. From the perspective of optimization, we should care more about the latter."*
>
> 	We compute the alignment between single-example updates and average these values over a mini-batch, following previous works (Appendix D; Lillicrap et al., Nature Communications 2016; Launay et al., arXiv:1906.04554 2019). We agree with the reviewer's comment on optimization and discussed this in the paper (lines 150–154).
>
> - *"In Fig. 2, could you comment on why the angle curve of the last readout (FC) layer seems to be diverging on fully connected VNNs? Also, why do the curves of the last readout (FC) layer first increase and then decrease rather than directly decrease on both fully-connected and convolutional VNNs?"*
>
> 	We failed to state an important aspect of Fig. 2, namely, that the last layer displayed in each plot is the penultimate layer and not the readout layer. The weight updates to the readout layer are by definition the same for GEVB, DFA, and BP. We will clarify this in the caption of Fig. 2 of the revised paper. Thus, to the first question, it is the alignment angle for the penultimate layer that appears to grow for fully connected VNNs. To interpret this observation, note that the penultimate-layer alignment angle is given by $\theta^{L-1} = \tan^{-1} r^{L-1}$ where $r^{L-1}$ is the relative standard deviation of the readout weights. Thus, the relative standard deviation of the readout weights is growing. While we are not certain why this occurs, this is likely harmless since, after 190 iterations, the alignment angle is well under 90 degrees ($\sim$60 degrees). To the second question, we note that nonmonotonic behavior of the alignment angles under DFA training has previously been observed and studied mathematically (Refinetti et al., arXiv:2011.12428 2020).
>
> - *"The current method seems to be surprisingly slow, e.g., taking 10 days of GPU training even on CIFAR and MNIST. Could you comment on what are the exact factors that slow down the training?"*
>
> 	We trained 48 different configurations of the form (MNIST/CIFAR-10, FC/CNN/LC, scalar/vector, sign-constrained/unconstrained, \{GEVB/DFA\}/BP) with five initializations each, resulting in 240 total trained models. Thus, training each model took one hour on average. The slowest models to train were vectorized locally connected CIFAR-10 models, which take significantly more than one hour to train. Prior work has noted that training of locally connected networks is computationally expensive (Bartunov et al., NeurIPS 2018).

---

### Author Response · Authors · 2021-08-10
**Collective response (see also responses to individual reviewers)**

We thank the reviewers for their constructive feedback. Reviewer osc2 believes that "this is a novel and interesting contribution towards more biologically plausible learning methods." Reviewer oN9Y states that our paper "proposes a very novel and intriguing way to approximate backprop." Reviewer Pci4 believes we have proposed "...an interesting alternative approach [to credit assignment] based on broadcasting an error signal..." Finally, reviewer wVMT notes that our work "...makes an interesting effort at providing `local' learning rules, which is an important limitation of BP; as such, the learning rule is novel."

Our paper presented a biologically plausible solution to the credit assignment problem in deep networks based on broadcasting a global error vector and applying local weight updates. The success of our proposed learning rule relies on a crucial theoretical result, namely, that the weight updates it produces are matched in sign to the gradient of a global loss function.

Here, we respond to a request shared by reviewers osc2, Pci4, and wVMT, namely, further detail regarding the biological implementation of vectorized nonnegative networks (VNNs) and the GEVB learning rule. In Section 7 of the paper, we described two possible biological implementations. In the revised paper, we will substantially expand this discussion so that the postulated neural circuit architectures and relationship to three-factor Hebbian learning are highly explicit. Additionally, we will describe distinct, testable predictions made by each implementation regarding the structure of the neuromodulatory signal. Finally, we will include a figure containing visual representations of these biological implementations. We summarize this expanded discussion below.

We will first consider an implementation in which each vector unit is implemented by a group of $K$ neurons.

**Architecture:** Consider a presynaptic group $j$ in layer $\ell - 1$ and a postsynaptic group $i$ in layer $\ell$. Neuron $\mu$ in group $j$ projects only to neuron $\mu$ in group $i$. Let $w^{\ell}\_{ij \mu}$ denote the strength of this connection. As vectorized weight sharing is not enforced automatically in neural circuits, this weight has a $\mu$ index.

**Nonlinearity:** The nonlinearity silences a group of neurons if the activity pattern of the group has positive alignment with the gating vector $\mathbf{t}$. This could be implemented through an inhibitory circuit mechanism within each group.

**Three-factor learning:** Given an input example and a target output, the network computes the error vector $\mathbf{e}$. The $K$ components of this error vector are then distributed throughout the network via $K$ distinct neuromodulatory signals, where the $\mu$-th signal is proportional to $e\_{\mu}$. Crucially, the $\mu$-th signal modulates only the $\mu$-th synaptic weight $w^{\ell}\_{ij \mu}$. Then, each synapse undergoes a three-factor Hebbian update given by $\Delta w^{\ell}\_{ij \mu} = - e\_{\mu} a^{\ell - 1}\_{j \mu} G(\mathbf{h}^{\ell}\_i)$. Here, $e\_{\mu}$ is the global third factor, $a^{\ell - 1}\_{j \mu}$ is the presynaptic firing rate, and $G(\mathbf{h}^{\ell}\_i)$ is a nonlinear function of the postsynaptic firing rate that is zero when the postsynaptic neuron is inactive and one when it is active. We recover the GEVB learning rule by relaxing the strengths of the $K$ updated synapses between groups $j$ and $i$ to their average, $\frac{1}{K}\sum\_{\mu} w^{\ell}\_{ij \mu}$. This relaxation to the average can be performed during an offline "sleep" phase. In particular, recent work demonstrates that lateral connections and anti-Hebbain plasticity can induce such a relaxation to the average (Pogodin et al., arXiv:2106.13031 2021). Future work could address how often this weight-tying operation must be performed to achieve effective learning.

Next, we will consider an implementation in which vectorization unfolds in time.

**Architecture:** Each vector unit is implemented by a single neuron. In each of $K$ consecutive time bins, a different set of input neurons are active, producing a single component of the network output. As the network processes inputs at different times using the same weights, vectorized weight sharing is enforced automatically.

**Nonlinearity:** As per the nonlinearity, each neuron must be active or inactive at all time bins according to the sign of $\mathbf{t} \cdot \mathbf{h}$, where $\mathbf{h}$ is the pre-activation vector, each element of which corresponds to a single time bin. Note that computing this inner product requires the full temporal input, violating causality. One workaround is to make $\mathbf{t}$ sparse in all but its first component so that each neuron is active or inactive based on its input at the first time step. Alternatively, gating could be implemented by an external inhibitory signal that is independent of the pre-activations. This technique is used in gated linear networks (Veness et al., arXiv:1910.01526 2019).

**Three-factor learning:** In time bin $\mu$, each synapse $w^{\ell}\_{ij}$ undergoes a three-factor Hebbian update given by $\Delta w^{\ell}\_{ij} = - e\_{\mu} a^{\ell - 1}\_{j \mu} G(\mathbf{h}^{\ell}\_i)$. Here, $e\_{\mu}$ is the global third factor at time bin $\mu$, $a^{\ell - 1}\_{j \mu}$ is the presynaptic firing rate at time bin $\mu$, and $G(\mathbf{h}^{\ell}\_i)$ is a nonlinear function of the postsynaptic firing rate that is zero when the postsynaptic unit is inactive and one when it is active for the current input spanning $K$ time bins. Integrating these updates over all $K$ time bins performs a sum over $\mu$, implementing the GEVB rule.

**Experimental predictions:** These biological implementations make distinct experimental predictions regarding the spatiotemporal structure of the neuromodulatory signal. The "grouped" implementation predicts that this signal is spatially heterogeneous, so that each synapse is modulated by the appropriate component of the error vector, and temporally uniform. By contrast, the "temporal" implementation predicts that this signal is temporally heterogeneous, so that the appropriate component of the error vector is broadcast at each time bin, and spatially uniform. Both implementations predict an important role for inhibition in silencing groups of neurons (grouped implementation) or single neurons (temporal implementation) in a stimulus-dependent manner.

---

### Decision · Program_Chairs · 2021-09-27

**Decision:**

Accept (Poster)

**Comment:**

The paper introduces a method to train neural networks in a way that is more biologically plausible than backpropagation. The key idea is to derive an update that depends on a global error signal. Work in this direction had already been identified for neural network with a scalar output, however the authors generalize the setup with vector outputs, but restricting the architectures to a very particular class of neural networks they introduce called VNN (effectively operating on vectors with vectorized weight sharing). They show the resulting update matches the sign of the gradient. Reviewers were in agreement the idea was interesting and elegant.
Doubts remained regarding biological plausibility, with reviewers thinking the authors arguments in that direction were somewhat hand-wavy. Nevertheless, we hope this work can inspire further work in that field that could be even better substantiated.
There were some concerns regarding the extensiveness of the literature review that should be addressed for the final version.